**PMIF v1.0: an inversion system to estimate the potential of satellite observations to monitor fossil fuel CO₂ emissions over the globe**

Yilong Wang[1,2,*], Grégoire Broquet[1], François-Marie Bréon[1], Franck Lespinas[1,3], Michael Buchwitz[4], Maximilian Reuter[4], Yasjka Meijer[5], Armin Loescher[5], Greet Janssens-Maenhout[6], Bo Zheng[1], Philippe Ciais[1]

[1]Laboratoire des Sciences du Climat et de l'Environnement, CEA-CNRS-UVSQ- Université Paris Saclay, 91191, Gif-sur-Yvette CEDEX, France
[2]The Key Laboratory of Land Surface Pattern and Simulation, Institute of Geographical Sciences and Natural Resources Research, Chinese Academy of Sciences, Beijing, China
[3]Canadian Centre for Meteorological and Environmental Prediction, 2121 Transcanada Highway, Dorval, QC, H9P 1J3, Canada
[4]Institute of Environmental Physics (IUP), University of Bremen FB1, Otto Hahn Allee 1, 28334 Bremen, Germany
[5]European Space Agency (ESA), Noordwijk, Netherlands
[6]European Commission, Joint Research Centre, Directorate Sustainable Resources, via E. Fermi 2749 (T.P. 123), I- 21027 Ispra, Italy

[*]*Correspondence to*: Yilong Wang (wangyil@igsnrr.ac.cn)

**Abstract.** This study assesses the potential of satellite imagery of vertically integrated columns of dry-air mole fractions of CO₂ (XCO₂) to constrain the emissions from cities and power plants (called emission clumps) over the whole globe during one year. The imagery is simulated for one imager of the Copernicus mission on Anthropogenic Carbon Dioxide Monitoring (CO2M) planned by the European Space Agency and the European Commission. The width of the swath of the CO2M instruments is about 300 km and the ground horizontal resolution is about 2 km resolution. A Plume Monitoring Inversion Framework (PMIF) is developed, relying on a Gaussian plume model to simulate the XCO₂ plumes of each emission clump and on a combination of overlapping assimilation windows to solve for the inversion problem. The inversion solves for the 3 h mean emissions (during 8:30-11:30 local time) before satellite overpasses and for the mean emissions during other hours of the day (over the aggregation between 0:00-8:30 and 11:30-0:00) for each clump and for the 366 days of the year. Our analysis focuses on the derivation of the uncertainty in the inversion estimates (the "posterior uncertainty") of the clump emissions. A comparison of the results obtained with PMIF and those from a previous study using a complex 3-D Eulerian transport model for a single city (Paris) shows that the PMIF system provides the correct order of magnitude for the uncertainty reduction of emission estimates (i.e. the relative difference between the prior and posterior uncertainties). Beyond the one or few large cities studied by previous studies, our results provide, for the first time, the global statistics of the uncertainty reduction of emissions for the full range of global clumps (differing in emission rate and spread, and distance from other major clumps) and meteorological conditions. We show that only the clumps with an annual emission budget higher than 2 MtC per year can potentially have their emissions between 8:30 and 11:30 constrained with a posterior uncertainty smaller than 20% for more than 10 times within one year (ignoring the potential to cross or extrapolate information between 8:30-11:30 time windows on

different days). The PMIF inversion results are also aggregated in time to investigate the potential of CO2M observations to constrain daily and annual emissions, relying on the extrapolation of information obtained for 8:30-11:30 time windows during days when clouds and aerosols do not mask the plumes, based on various assumptions regarding the temporal auto-correlations of the uncertainties in the emission estimates that are used as a prior knowledge in the Bayesian framework of PMIF. We show that the posterior uncertainties of daily and annual emissions are highly dependent on these temporal auto-correlations, stressing the need of systematic assessment of the sources of uncertainty in the spatiotemporally-resolved emission inventories used as prior estimates in the inversions. We highlight the difficulty to constrain global and national fossil fuel $CO_2$ emissions with satellite $XCO_2$ measurements only, and calls for integrated inversion systems that exploit multiple types of measurements.

## 1 Introduction

Cities, thermal power plants and industrial factories cover a very small fraction of the land surface but are emitting a large amount of $CO_2$. Many cities and regions are taking actions to reduce their greenhouse gas emissions. However, there are large uncertainties in the estimate of emissions from these $CO_2$ hotspots (Gately and Hutyra, 2017; Gurney et al., 2016). In addition, emissions at high temporal resolution (e.g. daily and hourly) depend on socio-economic activity and climate fluctuations, and thus have large variability. The large uncertainties and fluctuations of emissions at local scale have raised a growing political and scientific interest for an accurate and continuous monitoring of these local $CO_2$ emissions based on atmospheric measurements (Duren and Miller, 2012).

Measurements of $CO_2$ mole fractions from in situ surface networks, aircraft campaigns and mobile platforms around cities (Bréon et al., 2015; Lauvaux et al., 2016; Staufer et al., 2016) have been used to characterize the $CO_2$ signals downwind large cities and to quantify the underlying emissions based on an atmospheric inversion approach. However, such urban networks are deployed for few cities only. Alternatively, vertically integrated columns of dry-air mole fractions of $CO_2$ ($XCO_2$) from satellites offer the opportunity to sample the atmosphere with a global coverage. Kort et al. (2012) and Janardanan (2016) found that significant $XCO_2$ enhancements could be detected over some megacities using Greenhouse Gases Observing Satellite (GOSAT) $XCO_2$ observations. Schwandner et al. (2017) also found $XCO_2$ enhancements of 4.4 to 6.1 ppm in the Los Angeles urban $CO_2$ dome using observations from Orbiting Carbon Observatory-2 (OCO-2). Nassar et al. (2017) used the $XCO_2$ observations from OCO-2 to quantify $CO_2$ emissions from several middle- to large-sized coal power plants. However, the design of GOSAT and OCO-2 observations with sparse sampling was mainly focused on the monitoring of $CO_2$ natural fluxes. Recent studies show a limited amount of clear detections of transects of $XCO_2$ plumes from cities or plants in OCO-2 observations (Zheng et al., 2020) so that GOSAT and OCO-2 data keep on being hardly used to estimate $CO_2$ city emissions. The potential for reducing uncertainties in fossil fuel $CO_2$ emissions at the scale of point sources (Bovensmann et al., 2010), cities (Broquet et al., 2018; Pillai et al., 2016) and agglomerations of several cities (O'Brien et al., 2016) should dramatically change with the planned satellite missions with imaging capabilities. These studies consistently showed that imaging capability

with a wide swath (typically on the order of 200km – 300 km), a high resolution (< 2–3 km horizontal resolution) and a high single sounding precision (< 2 ppm) are required for satellite $XCO_2$ measurements for the monitoring of fossil fuel $CO_2$ emissions from large point sources and cities. Several satellite $XCO_2$ imagery concepts have been proposed: i) the OCO-3 NASA (National Aeronautics and Space Administration) mission which has been installed on the International Space Station (ISS) in May 2019; ii) the CarbonSat mission which was a candidate for ESA's Earth Explorer 8 opportunity (ESA, 2015), but was not selected; iii) the "city-mode" of the MicroCarb mission of the Centre National d'Etudes Spatiales (CNES) which should be launched in 2021 (Bertaux et al., 2019); iv) the GeoCARB geostationary mission which was selected as the Earth Venture Mission-2 by NASA; and v) the Copernicus Anthropogenic Carbon Dioxide Monitoring (CO2M) mission consisting of a constellation of $CO_2$ imagers that is currently studied by the European Space Agency (ESA) on behalf of the European Commission in the context of the European Union Copernicus programme. This CO2M satellite constellation is a crucial element that will contribute to the operational anthropogenic CO2 monitoring & verification support capacity currently under development by the European Commission with the support from ESA, European Organisation for the Exploitation of Meteorological Satellites (EUMETSAT) and the European Centre for Medium-Range Weather Forecasts (ECMWF) (Ciais et al., 2015; Pinty et al., 2017, 2019).

The main approach currently investigated for the estimate of $CO_2$ emissions from satellite $XCO_2$ images consists in identifying the $XCO_2$ plumes downwind the main $CO_2$ emission sources. The size of the plumes and the magnitude of $XCO_2$ enhancements in these plumes are tightly linked to the emissions. Wang et al. (2019) developed an algorithm to extract, from gridded emission maps, a conservative set of area (cities) and point sources (power plants) with intense emissions around the globe which can generate coherent $XCO_2$ plumes that may be observed from space, given the precision of current satellite observations. This set was conservative because it is inferred for idealized meteorological condition without wind. These emitting sources were called "emission clumps". Wang et al. (2019) identified 11,314 individual clumps which contribute 72% of the global fossil fuel $CO_2$ emissions from the ODIAC (Open-source Data Inventory for Anthropogenic $CO_2$ version 2017, Oda et al., 2018) 1 km resolution inventory.

Broquet et al. (2018) showed that the part of the $XCO_2$ plumes exploited by the atmospheric inversion in satellite images correspond to few hours of the clump emissions before the satellite overpass. The $XCO_2$ signature of the earlier clump emissions is too diluted to be filtered from the measurement errors and the signature of other $CO_2$ sources and sinks. Further, emissions from a given clump vary in time during the day, for instance due to the variations of traffic in cities (Yang et al., 2019), from day to day and between seasons, with more emissions associated to heating in winter over cold regions (Bréon et al., 2017). Therefore, the estimate of annual budgets of the clump emissions based on satellite observation during daytime (generally for a fixed local time since most of the missions use heliosynchronous orbits) and for low cloud coverage is a challenge, and cannot rely on the direct information from the satellite imagery. It relies on the extrapolation of information from the time windows for which the emissions are well constrained. Such an extrapolation is based on the correlation of the uncertainty in emissions in time, and more precisely, in the atmospheric inversion framework, on the temporal auto-correlations

of the uncertainty in the inventories used as a prior knowledge by the Bayesian framework of the inversion (see Sect. 2.6).

Previous studies on the potential of the satellite $XCO_2$ imagery to constrain the emissions from clumps were limited to single or few large targets, such as power plants in Bovensmann et al. (2010), Berlin in Pillai et al. (2016) and in Kuhlmann et al. (2019), and Paris in Broquet et al. (2018). However, much of the global $CO_2$ emissions occur in smaller cities and plants.
The potential and design of satellite missions dedicated to the monitoring of the $CO_2$ emissions like CO2M needs to be assessed

for a much more representative range of sources over the whole globe. The inversion framework used by Pillai et al. (2016) and Broquet et al. (2018) were based on a full 3-D Eulerian atmospheric transport models at high spatial resolution (on the order of 2 km). Such inversions are much too expensive in terms of computation cost, to be applied in a systematic way to the full set of clumps across the globe.

Therefore, in this study, we develop a Plume Monitoring Inversion Framework (PMIF) and conduct a set of Observing

System Simulation Experiments (OSSEs) to assess, for the first time, the performance of a satellite instrument to monitor the emissions of all the clumps across the globe and over a whole year. The imager studied has the foreseen characteristics of the individual satellites of the forthcoming CO2M mission. It would be a high-resolution spectrometer, with 2 km × 2 km resolution pixels and a swath of 300 km, and it would be placed on a sun-synchronous orbit ensuring global coverage in 4 days. The PMIF inversion system relies on the list of clumps extracted by Wang et al. (2019) from the ODIAC inventory, on the Gaussian

plume model to simulate the $XCO_2$ plumes generated by the emissions from these clumps, on an analytical inverse modeling framework, and on a combination of overlapping assimilation windows to solve for the inversion problem over the globe and a full year. It also addresses the question of temporal extrapolation that is needed to generate estimates of annual emissions from the information of a limited number of time windows for which emissions are well constrained by the direct satellite images, by accounting for the temporal auto-correlation of the prior uncertainties. The performance is assessed in terms of the

uncertainties in the emissions (Sect. 2.1) at different scales. The PMIF uses a Gaussian plume model at the local scale to ensure that the computation cost is affordable. Such a model can often hardly fit with actual plumes over the distances considered in this study (due to variations in the wind field, topography, vertical mixing etc. over such distances) but is shown, when driven with suitable parameters, to provide a satisfactory simulation of the plume extent and amplitudes, which appear to be the main drivers of the targeted computations of uncertainties in the emissions in our OSSE framework (as shown in section 3.1). In

PMIF, we also ignore the impact of some sources of uncertainties on the inversion of emissions, including systematic errors on the $XCO_2$ retrievals, the impact of uncertainties in diffuse anthropogenic emissions outside clumps, in non-fossil $CO_2$ fluxes (within and outside clumps), and in the spatial and temporal variations of emissions within the clump and the short time windows that the inversion aims to solve. These impacts are discussed in detail afterwards.

This PMIF system provides indication on the satellite system capabilities for the full range of cities and power plants

varying in topography, emission budget and spread, proximity to other major sources, and for a large range of meteorological conditions. It complements other systems that focus on specific regions with more complex (but area-limited) models and consideration of diffuse sources and natural fluxes, allowing for extrapolating and up-scaling results of those more complex

systems to get a more systematic understanding of their implications for the monitoring of $CO_2$ emissions from all detectible clumps over the globe.

The PMIF system and the OSSEs analyzed in this first study are described in Section 2. The results obtained with the PMIF for the city of Paris is compared with that of Broquet et al. (2018) in Sect. 3.1. The uncertainty in the retrieved emissions of individual clumps with one imaging satellite for 3 h time windows, for daily emissions and for annual emissions are assessed in Sect. 3.2-3.4. Sect. 4 discusses the drivers of the spatial variations of the uncertainty in the retrieved emissions, the limitations of PMIF, and the implications for a future operational observing system.

## 2. Methodology

### 2.1 Plume Monitoring Inversion Framework

The theoretical framework of the inversion system developed in this study is the same as the traditional atmospheric inversions. The inversion derives a statistical estimate for a set of control variables $x$ in a model $x{\rightarrow}y=\mathbf{M}x$ that simulates the satellite $XCO_2$ measurements $y^o$. The model $\mathbf{M}$ linking $x$ and $y$ is a combination of flux and atmospheric transport models (detailed in Sect. 2.4), and is called observation operator hereafter. As explained below, we do not have a constant term added to $\mathbf{M}x$ in the observation operator of the PMIF that would gather the atmospheric $CO_2$ signature of the fluxes not controlled by the inversion (like non-fossil fluxes and the background $XCO_2$ field) since the uncertainty in such fluxes is ignored. The inversion follows a Bayesian statistical framework, updating the statistical prior estimate of $x$ based on the statistical information from the assimilation of $XCO_2$ measurements $y$ into the observation operator. The distributions of the prior estimate and of the misfits between the actual observations $y^o$ and simulated ones due to errors in the observations and in the observation operator (called the "observation errors") are assumed to be unbiased and to have the Gaussian forms $N(x^b, \mathbf{B})$ and $N(0, \mathbf{R})$, where $\mathbf{B}$ and $\mathbf{R}$ are the prior and observation error covariance matrices. The statistical distribution of the posterior estimate of $x$, given the observation operator, $x^b$ and $y^o$, also follows a Gaussian distribution $N(x^a, \mathbf{A})$, with $x^a$ being the mean and $\mathbf{A}$ being the error covariance matrix characterizing the posterior uncertainty. The problem is solved by deriving:

$$\mathbf{A} = (\ \mathbf{B}^{-1} + \mathbf{M}^T\mathbf{R}^{-1}\mathbf{M}\ )^{-1} \tag{1}$$

$$x^a = x^b + \mathbf{A}\mathbf{M}^T\mathbf{R}^{-1}\ (\ y^o - \mathbf{M}x^b - y^{\text{fixed}}\ ) \tag{2}$$

Where $^T$ and $^{-1}$ denote the transpose and inverse of a given matrix.

Equation (1) shows that $\mathbf{A}$ only depends on prior and observation error covariance matrices, on the matrix part of the observation operator (hereafter, we simplify the notation by calling $\mathbf{M}$ the observation operator), and implicitly on the structure of the observation vector (i.e., on the time, location and representation of the observations in $\mathbf{M}$), while Eq. (2) shows that $x^a$ also depends on the actual value of $x^b$ and $y^o$. PMIF is an analytical inversion system that solves for Eq. (1) or for an approximation of this equation (when accounting for temporal correlations in $\mathbf{B}$) by building the different matrices involved

in this equation.

We characterize **B**, **R** and **A** by the corresponding standard deviations ($\sigma$) of uncertainty in individual or aggregations of control parameters and by the temporal auto-correlations of the uncertainties (Sect. 2.6). In the following, the "uncertainty reduction" for a given control variable or for an aggregation of control variables (like emission budgets over larger timescales than that of the control vector) refers to the relative difference between its prior and posterior uncertainty: $1 - \sigma_a/\sigma_b$.

We use a Gaussian plume model (Sect. 2.4) to simulate the atmospheric transport at a spatial resolution consistent with that of the $XCO_2$ measurements from the planned $CO_2$ imager and with the highly heterogeneous distribution of emissions. Compared with complex 3-D atmospheric transport models, Gaussian plume models have a very low computational cost, making the global assessment of posterior uncertainty and uncertainty reduction at the scale of emissions clumps from the assimilation of high resolution data feasible. However, since a Gaussian plume model provides a highly simplified approximation of the atmospheric transport from emission clumps, we need to verify that its use in the PMIF yields estimates of the uncertainties in the inverted emissions that are consistent with those that would be based on more complex models. Therefore, we first compare the results for Paris from PMIF against those acquired based on a 3-D Eulerian atmospheric transport model by Broquet et al. (2018), the latter also accounting for uncertainties in diffuse $CO_2$ fluxes. On the one hand, the signals from these diffuse and natural $CO_2$ fluxes cannot be modelled effectively by a Gaussian plume model. On the other hand, the diffuse and natural $CO_2$ fluxes in Paris was shown to have only a weak impact on the inversion of fossil fuel $CO_2$ emissions (Staufer et al., 2016). For this comparison, we use the same simulation of the $XCO_2$ sampling by CarbonSat (Sect. 2.2) and a similar control vector as Broquet et al. (2018). The corresponding inversion with the PMIF is called PMIF-Paris hereafter. Then we apply the system to all the emission clumps over the globe and over 1 year using a different control vector and a simulation of the $XCO_2$ sampling by a single CO2M satellite (Sect. 2.2). The inversions for all emission clumps over the globe are called PMIF-Globe. In PMIF-Globe, we first investigate the potential of satellite observations in constraining emissions from individual days (ExpNoCor in Sect. 2.6). Then we assess the ability of satellite observations to constrain emissions at annual scale by accounting for the temporal auto-correlation of the prior uncertainties (other experiments in Sect. 2.6). Table 1 and 2 summarize the different options for the configuration of the system and of the OSSEs. One distinction between PMIF-Paris and PMIF-Globe is that PMIF-Paris relates $XCO_2$ signals with the mean emissions 6 hours before overpasses, while it is assumed that in PMIF-Globe that the $XCO_2$ signals only provide effective constraints on 3 h mean emissions before individual overpasses. The 6-hour period corresponds to the period of emissions from Paris whose signature in the $XCO_2$ field can still be detected by the satellite despite the atmospheric diffusion (Broquet et al., 2018). While Broquet et al. (2018) indicated that the period of "detectable" emissions from a large megacity like Paris could last up to 6-hours, most of the clumps across the globe have smaller emission rates than Paris, or are located in more complex environment close to other major emission areas where $XCO_2$ signals can be attributed to multiple sources, making the detection of the $XCO_2$ signature of emissions few hours before the satellite overpass even more difficult. For the PMIF-Globe experiments, we thus conservatively assume that the $XCO_2$ signals can only provide effective constraints on 3 h mean emissions before individual

overpasses in general.

**Table 1** The configuration of PMIF-Paris inversion

| Type of setting | Option |
| --- | --- |
| Control vector | 6-hour mean fossil fuel $CO_2$ emissions from Paris over 5:00-11:00 (local time is used) |
| Plume length in the computation of **M** | 6 hour × wind speed averaged over 5:00-11:00 |
| Observation sampling and measurement error | Simulation of the sampling and random measurement noise for CarbonSat near Paris |
| Prior uncertainty | 22.4% for the 6-hour mean emissions<br>The potential correlations between the 6-hour mean emissions of different days are ignored for the diagnostics |

**Table 2** The different options for the configuration of PMIF-Globe inversions

| Type of setting | Option |
| --- | --- |
| Control vector | For each clump of the globe, 3-hour mean emissions over 8:30-11:30 and the mean emissions for the remaining 21 hours (0:00-8:30 plus 11:30-24:00) within each day of 1 year |
| Plume length in the computation of **M** | 3 hour × wind speed averaged over 8:30-11:30; no computation of plume for the emissions over 0:00-8:30 plus 11:30-24:00 |
| Observation sampling and measurement error | Simulation of the sampling and random measurement noise for a single CO2M $CO_2$ imager all over the globe |
| Constraint on the prior uncertainty | For each clump, the budget of the prior uncertainty in annual emission is 30%. The uncertainty in the 3 h mean emissions and in the budget of the emissions for the rest of the day are downscaled depending on the assumptions on the components of the prior uncertainty and on their temporal auto-correlations (see Sect. 2.6) |

## 2.2 Observation space

In this study, we consider the samplings from two different virtual $CO_2$ imagers.

The first sampling used in PMIF-Paris (Table 1 and Sect. 2.7.1) is the simulation of the sampling for CarbonSat by Buchwitz et al. (2013) exactly as in Broquet et al. (2018). $XCO_2$ is sampled by a 240 km swath instrument with 2 km spatial resolution. Given the presence of cloud and aerosol and their impacts on the precision of $XCO_2$ retrievals, only "good" $XCO_2$ observations, for which the sum of the retrieved aerosol optical depth (AOD) at NIR wavelength and atmosphere cirrus optical depth (COD) is less than 0.3, are used in the inversions. The preferable condition, AOD(NIR)+COD<0.3, for a good $XCO_2$

observation is referred to as "clear sky" hereafter. The CarbonSat sampling was simulated over the whole globe and for a full year by Buchwitz et al. (2013), but it is used here for the inversion of the emission of Paris only. Thus, only the passes with at least one good $XCO_2$ measurement in the 100km radius circle centered on Paris are used, as in Broquet et al. (2018).

    The second sampling is global and is used for all the other experiments of PMIF-Globe (Table 2 and Sect. 2.7.2). It corresponds to that of a single CO2M satellite with a 300 km swath and 2 km spatial resolution. CO2M is similar to CarbonSat

for sampling, but has a larger swath, and a better precision (Sect. 2.5). The simulation is based on the method and model described by Buchwitz et al. (2013), but uses different values for the parameters in the model.

## 2.3 Control vector

    In the PMIF-Paris inversion, the satellite observations are sampled at 11:00 local time, in line with the experiments from Broquet et al. (2018). The inversion solves for the mean emissions for the 6 hours before 11:00 local time. Broquet et al. (2018)

solved for the hourly emissions during this 6-hour period but PMIF can only solve for the mean emissions during the 6-hour period due to the fact that the Gaussian plume model cannot be used to compute the signatures in the $XCO_2$ field of individual hourly emissions during that period. The control parameter in PMIF-Paris for each overpass (Sect. 2.7.1) is thus a scaling factor $\lambda$ for the mean emission between 05:00 and 11:00. The prior and posterior scaling factors are used to rescale the 1 h and ~1 km resolution emission fields from an emission map and its temporal profile which are parts of the observation operator

(Sect. 2.4).

    In the PMIF-Globe inversion, the satellite observations are sampled at a local time of approximately 11:30 over all the clumps. The inversion solves for a scaling factor for 3-hour mean emissions between 8:30 and 11:30 and a scaling factor for the emissions during of the rest of the day (0:00-8:30 plus 11:30-24:00) for each day over one year and for all the clumps over the globe:

$$x=[\lambda_{clump1}^{day1,morning}, \lambda_{clump1}^{day1,rest}, \lambda_{clump1}^{day2,morning}, \lambda_{clump1}^{day2,rest}, \ldots, \lambda_{clump1}^{day366,morning}, \lambda_{clump1}^{day366,rest}, \lambda_{clump2}^{day1,morning},$$
$$\lambda_{clump2}^{day1,rest}, \ldots \lambda_{clumpN}^{day366,morning}, \lambda_{clumpN}^{day366,rest}]$$
(3)

    While Broquet et al. (2018) indicated that the period of "detectable" emissions from a large megacity like Paris could last up to 6-hours, most of the clumps across the globe have smaller emission rates than Paris, or are located in more complex environment close to other major emission areas where $XCO_2$ signals can be attributed to multiple sources, making the

detection of the $XCO_2$ signature of emissions few hours before the satellite overpass more difficult. For the experiments other than PMIF-Paris, we thus conservatively assume that the $XCO_2$ signals can only provide effective constraints on 3 h mean

emissions before individual overpasses in general, and we use the 8:30-11:30 time window for all emission clumps over the globe. The control vector is defined using this time window for all the days of the year, and not only for the days with satellite local overpasses, to facilitate the definition of the prior uncertainties and the combination of results at the annual scale. .

In both types of experiments, we do not include the diffuse emissions outside the selected clumps and the natural fluxes (more generally, any parameter of the "background concentrations", Kuhlmann et al., 2019) in the control vector. The set-up of the **R** matrix also ignores uncertainties in the background concentrations (Sect. 2.5). This is another divergence with the inversion configuration of Broquet et al. (2018) who accounted for such uncertainties.

## 2.4 Observation operator

The observation operator in PMIF (which is used in Eq. 1) is composed of two sub-operators. The first operator ($\mathbf{M}_{\text{inventory}}$) describes the spatial distribution (within the clumps) and temporal variations of the emissions whose budgets are controlled by the inversion during 8:30-11:30 and during the remaining 21 hours for each clump: $x \rightarrow E = \mathbf{M}_{\text{inventory}}x$. The spatial distribution of the emissions are based on estimates from ODIAC (Oda et al., 2018) for the year 2016. ODIAC provides the monthly mean emissions for 12 months through a year at a 0.0083°×0.0083° (approximately 1 km×1 km) spatial resolution. The weekly and diurnal (at hourly resolution) profiles from the Temporal Improvements for Modeling Emissions by Scaling (TIMES) product (Nassar et al., 2013) are applied to the monthly emission maps of ODIAC to generate the hourly emission fields. The second operator ($\mathbf{M}_{\text{plume}}$) simulates the plumes of $XCO_2$ enhancement above the background at and downwind the emission clumps at 11:30: $E \rightarrow y = \mathbf{M}_{\text{plume}}E$. We assume that the plume of $XCO_2$ enhancement related to a given emitting pixel within a clump of the ODIAC map has a Gaussian shape and the plume from a clump is a sum of multiple Gaussian plumes from all the ODIAC pixels within that clump. For a given emitting pixel, the Gaussian plume model writes:

$$y(i,j) = \alpha \frac{E}{\sqrt{2\pi}\sigma_j u} e^{-\frac{j^2}{2\sigma_j^2}} \tag{4}$$

Where $y$ is the $XCO_2$ enhancement (in ppm) downwind of the emitting pixel. The $i$-direction is parallel to the wind direction and the $j$-direction is perpendicular to the wind direction. $y$ depends on the mean emission rate during 8:30-11:30 at local time (E, in g/s), the wind speed ($u$, in m/s), the cross-wind distance ($j$) and the parameter $\sigma_j$ (see below). The wind direction and speed is taken from the Cross-Calibrated Multi-Platform (CCMP) gridded surface wind fields for the year 2008 (Atlas et al., 2011). The CCMP product uses a Variational Analysis Method (VAM) to combine the data from Version-7 RSS radiometer wind speeds, QuikSCAT and ASCAT scatterometer wind vectors, moored buoy wind data, and ERA-Interim model wind fields. The $\sigma_j$ is a function of downwind distance $i$ and atmospheric stability parameter: $\sigma_j = \beta j/(1+\gamma j)^{-1/2}$, where $\alpha$ is a coefficient that converts the computed $XCO_2$ enhancement in the unit of ppm, and $\beta$ and $\gamma$ are coefficients depending on the atmospheric Pasquill stability category which is a function of the wind speed and solar radiation (Turner, 1970). The values for $\beta$ and $\gamma$ can be found in Bowers et al. (1980). The original Gaussian plume model generates a stationary plume of an infinite length and width downwind the emissions. Because we assume that the $XCO_2$ plumes sampled from a satellite overpass is only related to

the emissions 3 h before, the Gaussian plume corresponding to each emitting pixel is cut off at the downwind distance equaling the wind speed multiplied by 3 h. The width of the plume is also cut off beyond 3 times of $\sigma_j$ in the cross-wind direction. The observation operator is null for emission of the remaining 21 hours (0:00-8:30 plus 11:30-24:00).

The size of the full theoretical control vector corresponds to 11,314 emission clumps times two time windows for each day times 366 days. The size of this full theoretical observation vector over the year is thus more than 30,000,000. Building matrices and applying Eq. 1 with such spaces is, in practice, not computationally affordable. Therefore, we divide the globe into 5,400 spatial inversion windows (from 180ºW to 180ºE and from 90ºN to 60ºS), each inversion window covering an area of 10º×10º and being extended on the four boundaries with margins of 500 km to ensure that the plumes from the clumps near the boundary of inversion windows are fully simulated and accounted for in the corresponding inversions. $\mathbf{M_{plume}}$ is defined as a block matrix, each block representing a single spatial inversion window and a single day. When an emission clump and its plume are comprised within more than one inversion window on a single day, only the results obtained in the window that covers the full plume is used in $\mathbf{M_{plume}}$.

## 2.5 Observation error

We evaluate the projection of the measurement noise of the satellite observation, and ignore uncertainties in the observation operator. The measurement noise is derived from the simulations of random measurement errors from Buchwitz et al. (2013) and the impact of the systematic measurement errors is ignored. The random measurement errors are simulated as a function of geographic location (e.g., solar zenith angle, SZA), surface (e.g. albedo) and atmosphere characteristics (e.g. aerosol optical depth, AOD). The random measurement error is 1.4 ppm for vegetation albedo and SZA 50º in the CS sampling, and it is 0.7 ppm in the CO2M sampling, thus two-fold smaller for the latter. The random measurement errors are uncorrelated from one $XCO_2$ data to the other, and the $\mathbf{R}$ matrix is thus built as a diagonal matrix as generally done in atmospheric inversion.

## 2.6 Specification of the prior uncertainties and of their temporal auto-correlations

Two configurations for the prior uncertainty are used in the OSSEs (Sect. 2.7). In the PMIF-Paris inversion, the prior uncertainty is 22.4% for the 6-hour mean emissions, the choice of this value being consistent with the configuration used by Broquet et al. (2018).

In the PMIF-Globe inversions, the prior uncertainty is downscaled from its estimate for the annual budget of emissions of each clump. A prior uncertainty in annual emission of 30% is assumed for all clumps. This value is chosen to be of the same order of magnitude as the typical difference between emission inventories for a single point source and city. For example, Gurney et al. (2016) found that one-fifth of the power plants had monthly emission differences larger than 13% between the estimates by two different US agencies. Gurney et al. (2019) compared the emission maps from ODIAC and Hestia for four US cities and found the whole-city differences are between -1.5% and +20.8%. Gately and Hutyra (2017) compared the inventories reported by local authorities and bottom-up fossil fuel $CO_2$ emission maps for 11 US cities and found the differences

range from 33% to 78%. Then, the downscaling of the uncertainty in annual emissions into uncertainties at the sub-daily scale of the control variables (i.e. 3 h mean emission over 8:30-11:30 and 21 h mean emission during the rest of the day; Sect. 2.3) follows a decomposition of the total uncertainty into components with different temporal auto-correlations.

The hourly emissions in inventories are usually derived from the periodic typical temporal profiles to annual emissions (Andres et al., 2011; Nassar et al., 2013). There are large variations in actual emissions from hour to hour and from day to day, resulting in large differences between the emission estimates derived based on typical temporal profiles and actual emissions. These differences are sources of uncertainties in the emission inventories which are used in the inversion as prior information. However, there is no consensus regarding the uncertainty in emission inventories and their error structures (Gurney et al., 2019). We compare the typical temporal profiles of transport emissions and energy sector from the TIMES product respectively with the TOMTOM traffic index (https://www.tomtom.com/en_gb/, that provides indications on the level of variability in the traffic even though not on that of the $CO_2$ emission themselves), and with the actual hourly $CO_2$ emissions from electricity production in France (https://www.services-rte.com/en/home.html). Although these comparisons are only made for two sectors, the results already show that it is challenging to describe the temporal auto-correlations of the uncertainty in emissions with simple exponentially decaying functions (Fig. S1 and S2) like what is usually done in traditional atmospheric inversions (Chevallier et al., 2010; Kountouris et al., 2015). We thus make several assumptions regarding the decomposition of the prior uncertainty into components with different modes of auto-correlation.

In some scenarios, we consider an "annual component" that is fully correlated in time over 1 year. We also consider "uncorrelated" components whose temporal auto-correlations are null and "sub-annual" components whose temporal auto-correlations follow the exponential decaying model with a correlation length smaller than 1 year. Specifically, we assume that the correlation between two instants of the sub-annual component at the hourly scale is described by:

$$r = \exp(-\Delta h / \tau_1) \times \exp(-\Delta d / \tau_2) \qquad (5)$$

Where $\Delta h$ is the time lag (in hours) between the two times of the day that are considered and $\Delta d$ is the time lag (in days) between the two dates that are considered. The parameters $\tau_1$ and $\tau_2$ follow the fit of the misfits between the TIMES profiles and the TOMTOM and electricity production indices to the exponential functions respectively at the hourly scale and at the daily scale (Fig. S1 and S2). The temporal auto-correlations between the emissions during the aggregated time windows (8:30-11:30 and the remaining 21 hours) are computed by re-aggregating the uncertainties at the hourly scale accounting for temporal auto-correlation.

The detailed configuration of the different scenarios for the decomposition of the prior uncertainty are listed below:

1) Annual component and Moderately correlated Sub-annual component (AMS): composed of an annual component and a sub-annual component. The temporal auto-correlation of the sub-annual component follows Eq. (5) with $\tau_1$=12h and $\tau_2$=7d. The ratio of the uncertainty in annual component to that in sub-annual component for 3 h emissions is assumed to be 3:5. This leads to an annual uncertainty component $\sim N(0, 29\%)$ and a sub-annual component $\sim N(0, 49\%)$ for 3 h emissions and $\sim N(0, 38\%)$ for 21 h emissions.

2) Annual component and Strongly correlated Sub-annual component (ASS): composed of an annual component and a sub-annual component. The temporal auto-correlation of the sub-annual component follows Eq. (5) with $\tau_1$=2400h, which approximately corresponds to having full correlations between hourly uncertainties within a single day, and $\tau_2$=20d. The ratio of the uncertainty in annual component to that in sub-annual component for 3 h emissions is assumed to be 3:5. This leads to an annual uncertainty component ~$N$(0, 26%) and a sub-annual component ~$N$(0, 44%) for 3 h emissions and ~$N$(0, 44%) for 21 h emissions.

3) Moderately Correlated Sub-annual component (MCS): composed of a sub-annual component. The temporal auto-correlation of the sub-annual component follows Eq. (5) with $\tau_1$=12h and $\tau_2$=7d. This leads to an sub-annual component ~$N$(0, 198%) for 3 h emissions and ~$N$(0, 119%) for 21 h emissions.

4) Strongly Correlated Sub-annual component (SCS): composed of a sub-annual component. The temporal auto-correlation of the sub-annual component follows Eq. (5) with $\tau_1$=2400h and $\tau_2$=20d. This leads to a sub-annual component ~$N$(0, 93%) for 3 h emissions and ~$N$(0, 93%) for 21 h emissions.

5) Sector-dependent Correlated Sub-annual component (SectCS): composed of a sub-annual component for each emission sector. It is assumed that the relative uncertainty for different sectors are the same. The temporal auto-correlation of the sub-annual components for all sectors follow the same formulation Eq. (5), but with different $\tau_1$ and $\tau_2$. For the emissions in the industry sector, $\tau_1$=2400h and $\tau_2$=180d; for the emissions in the transport sector, $\tau_1$=12h and $\tau_2$=7d; for the emissions from energy sector: $\tau_1$=24h and $\tau_2$=7d; and for the emissions from other sectors: $\tau_1$=24h and $\tau_2$=14d. For each clump, the share of emissions from each sector are estimated according to EDGARv4.3.2 (https://edgar.jrc.ec.europa.eu/). This leads to an uncertainty in 3 h emissions ranges between 40% and 198%, and in 21 h emissions ranges between 40% and 154%.

6) No temporal auto-correlation (NoCor): we assume that the uncertainties in 3 h emissions and 21 h emissions on all days are all random and uncorrelated from one time window to the other, or from one day to the other. The resulting sub-annual component follows the distribution ~$N$(0, 1623%) for 3 h emissions and ~$N$(0, 614%) for 21 h emissions.

The prior uncertainty in the 3-h mean emissions between 8:30 and 11:30 is close to or larger than 100% in scenarios SCS and MCS, and it even reaches an abnormally huge value of 1623% in NoCor. Andres et al. (2016) estimated the uncertainty in the widely used emission map CDIAC (Carbon Dioxide Information Analysis Center). They found that the average uncertainty in monthly emissions for one 1º×1º grid cell is 120% and further suspected that the uncertainties in hourly and daily emissions at urban scale could be even larger (from a few percent to 1000%). But these large values challenges the assumption that the uncertainty in anthropogenic emissions is normally distributed (Gurney et al., 2019). In this study, we follow the traditional assumption used in atmospheric inversions that the prior uncertainty follows a Gaussian distribution, allowing the prior uncertainty to exceed 100% in some scenarios. This assumption ensures that the system is analytically solvable using Eq. (1) and (2). In addition, we focus our analysis on 8:30-11:30 time windows or days for which the posterior uncertainties of underlying emissions are smaller than 20% (Sect. 2.7.2), a value that is significantly smaller than the prior uncertainty in any scenario. In these cases, Eq. (1) ensures that the posterior uncertainty is almost driven the projection of the observation error

on the control space and is not sensitive to the level of prior uncertainty.

## 2.7 Practical implementation of the OSSEs

Two sets of OSSEs are conducted under different configurations adapted to different purposes, as described below. Table 1 and 2 summarize the different configurations of the OSSEs.

### 2.7.1 Comparison of results between PMIF and a previous study on a single city: Paris

In the first OSSE PMIF-Paris, the configuration of the control vector, observation sampling and errors, and prior uncertainties are made such that they resemble those in the MC-2 experiments from Broquet et al. (2018): 1) the inversion controls the 6-h mean emissions from Paris before the satellite overpasses on single days; 2) the observation sampling and errors are obtained from CarbonSat mission simulation (Buchwitz et al., 2013); 3). We ignore temporal auto-correlation of the uncertainty in 6-h mean emissions between different days. We select the same 69 satellite CarbonSat overpasses over Paris

during one year as Broquet et al. (2018). The 31 days of October 2010 are used to provide a wide sample of atmospheric transport conditions. These atmospheric transport conditions are combined with the 69 sets of CarbonSat overpasses (with various cloud and aerosol coverage) to form 2139 inversion samples. The results for different overpasses on a single day are ranked according to the uncertainty reductions and are compared to those obtained in Broquet et al. (2018).

### 2.7.2 Applying the PMIF over all emission clumps across the globe

In this second set of OSSEs, PMIF-Globe, we conduct inversions for all the clumps over one year. However, the large sizes of the control vector, of the observation vector and of the associated covariance matrices prevent the derivation of a full **A** for all the clumps and all the time windows using Eq. (1). In PMIF, we thus propose and apply a two-step computation that approximates Eq. (1). This computation assumes that the system has a limited capability to improve the separation between plumes from distinct clumps on a given day by crossing the information obtained from different days. In that sense, the

inversion considers the uncertainty reduction obtained for individual days when considering all the clumps together (first step, see below) before focusing on individual clumps to account for temporal correlations in the prior uncertainty (the second step, see below). In other words, we assume that when crossing information between different time windows for a given clump, the impact of filtering information from different spatial overlaps of plumes on different days is relatively smaller than that of temporal auto-correlation in the prior uncertainty. It is proven that this method provides a good approximation of **A** at daily to

annual scales for individual clumps (Supplementary text S1).

In a first step, Eq. (1) is applied to each $10°×10°$ spatial inversion windows on each day separately (corresponding to an 8:30-11:30 time window for clumps within the spatial inversion windows), by using the corresponding blocks in **B**:

$$\mathbf{A}_{\text{spt,i,j}} = \left( \mathbf{B}_{\text{spt,i,j}}^{-1} + \mathbf{M}_{\text{spt,i,j}}^{\text{T}} \mathbf{R}_{\text{spt,i,j}}^{-1} \mathbf{M}_{\text{spt,i,j}} \right)^{-1} \tag{6}$$

Where $i$ is the $i$th spatial inversion window and $j$ is the $j$th day during one year. Here, $\mathbf{B}_{\text{spt,i,j}}$ is a diagonal matrix that only contains the variances of prior uncertainties in emissions during 8:30-11:30 for the clumps within the inversion window. $\mathbf{M}_{\text{spt,i,j}}$ accounts for the spatial overlap of plumes generated from nearby clumps. Then we derive a "instant" $\mathbf{M}^{\text{T}}\mathbf{R}^{-1}\mathbf{M}$ (denoted as $\mathbf{M}_{\text{I,J,k}}^{\text{T}}\widehat{\mathbf{R}_{\text{I,J,k}}^{-1}}\mathbf{M}_{\text{I,J,k}}$) for a given clump $k$ at each 8:30-11:30 time window:

$$\mathbf{M}_{\text{I,J,k}}^{\text{T}}\widehat{\mathbf{R}_{\text{I,J,k}}^{-1}}\mathbf{M}_{\text{I,J,k}}=\left(\mathbf{A}_{\text{spt,i,j}}(k)^{-1} - \mathbf{B}_{\text{spt,i,j}}(k)^{-1}\right)^{-1} \tag{7}$$

Where $a_{\text{spt,i,j}}(k)$ is a scalar from $\mathbf{A}_{\text{spt,i,j}}$ representing the variance of posterior uncertainty of emission from clump $k$ in $i$th spatial inversion window and in 8:30-11:30 time window on day $j$ obtained by Eq. (6), and $b_{\text{spt,i,j}}(k)$ is the scalar from $\mathbf{B}_{\text{spt,i,j}}$ representing the variance of prior uncertainty for the same control variable.

In the second step, the inversion is conducted for each clump $k$ separately, considering the correlation in time in $\mathbf{B}$, using $\mathbf{M}_{\text{I,J,k}}^{\text{T}}\widehat{\mathbf{R}_{\text{I,J,k}}^{-1}}\mathbf{M}_{\text{I,J,k}}$ derived from the first step:

$$\mathbf{A}_{tmp,\text{k}} = \left( \mathbf{B}_{tmp,\text{k}}^{-1} + \begin{bmatrix} \mathbf{M}_{\text{I,1,k}}^{\text{T}}\widehat{\mathbf{R}_{\text{I,1,k}}^{-1}}\mathbf{M}_{\text{I,1,k}} & 0 & 0 \\ 0 & \ddots & 0 \\ 0 & 0 & \mathbf{M}_{\text{I,n,k}}^{\text{T}}\widehat{\mathbf{R}_{\text{I,n,k}}^{-1}}\mathbf{M}_{\text{I,n,k}} \end{bmatrix} \right)^{-1} \tag{8}$$

Where n=366×2, representing the time windows for 8:30-11:30 and for the rest 21 hours on the 366 days of one year (2008). $\mathbf{B}_{\text{tmp,k}}$ is the covariance matrix accounting for the temporal auto-correlation in the prior uncertainty for a single clump:

$$\mathbf{B}_{tmp,k} = \begin{bmatrix} \sigma_{t1}^2 & cov(\varepsilon_{t1}, \varepsilon_{t2}) & \dots & cov(\varepsilon_{t1}, \varepsilon_{tn}) \\ cov(\varepsilon_{t1}, \varepsilon_{t2}) & \sigma_{t2}^2 & \dots & cov(\varepsilon_{t2}, \varepsilon_{tn}) \\ \vdots & \vdots & \ddots & \vdots \\ cov(\varepsilon_{t1}, \varepsilon_{tn}) & cov(\varepsilon_{t2}, \varepsilon_{tn}) & \dots & \sigma_{tn}^2 \end{bmatrix} \tag{9}$$

In PMIF-Globe, we first conduct the inversion in which the prior uncertainty has no temporal auto-correlation (Exp-NoCor). This is made by applying step 1 to all the 10°×10° spatial inversion windows and all the days separately. This case is used to label the "well constrained" 8:30-11:30 time windows for a given clump when the associated plume is sufficiently well sampled by the $XCO_2$ observation to yield a posterior uncertainty in the 3 h mean emission that is smaller than 20%. We then conduct inversions with different assumptions about the decomposition of the prior uncertainty, accounting for the impact of temporal auto-correlations of the prior uncertainty by applying step 2 of the inversions. The posterior uncertainties in the 3 h mean emissions labeled in Exp-NoCor are compared among different inversions to show the benefit of crossing information from different time windows. Apart from the assessment of the posterior uncertainties for the 3 h mean emissions, we also evaluate, for all the experiments except Exp-NoCor, the posterior uncertainty in daily emissions and in annual emissions by aggregating the posterior uncertainty covariance matrix $\mathbf{A}$ at the corresponding scales obtained in step 2 of the inversions.

## 3. Results

**3.1 Comparison between results from PMIF and a more complex but local system over an isolated megacity: Paris**

The comparison of the results from the PMIF-Paris experiment to that of Broquet et al. (2018) is used to demonstrate that the PMIF produce meaningful statistics for other clumps despite its relative simplicity at the local scale (its complexity being linked to its global and annual coverage). Figure 1 shows the theoretical uncertainty reduction for the 6 h mean emissions obtained in PMIF-Paris inversions with the 1st, 5th, 10th, 15th, 19th and 25th best observation sampling from CarbonSat over 31 inversion days (Sect. 2.7.1), each day being characterized by the average wind speed over Paris. We compare these results with the Fig. 6 from Broquet et al. (2018). Like Broquet et al. (2018), Fig. 1 illustrates the strong correlation between the uncertainty reduction and the average wind speed, indicating that lower wind speed results on a larger signal close to the city that is easier to assimilate than elongated plumes under large wind speeds. For the best observation sampling, the uncertainty reduction remains smaller than 40% when the wind speed is larger than 13 m s$^{-1}$, and this value is generally twice as low as the values obtained when the wind speed is smaller than 5 m s$^{-1}$.

Some differences are seen in Fig. S3, between the results obtained by PMIF and by Broquet et al. (2018). For example, the PMIF-Paris inversion slightly overestimates the uncertainty reduction under high wind speed (> 15 m s$^{-1}$) using the best observation sampling compared to Broquet et al. (2018). These differences reflect the impact of using the Gaussian plume model instead of a 3-D atmospheric transport model, and more importantly, the impact of accounting for more sources of uncertainties (in diffuse emissions and natural fluxes) in Broquet et al. (2018). Despite these differences, the general coherence in the ranges of uncertainty reductions (Fig. S3) under different wind speeds between the PMIF-Paris experiment and Broquet et al. (2018) is a strong indication that the PMIF generates the correct order of magnitude for the uncertainty reduction for a single clump. In addition, Nassar et al. (2017) used the Gaussian plume model to process actual XCO$_2$ plumes generated from several power plants, which were sampled by OCO-2, adding the indication that Gaussian plume model can simulate the typical spread and amplitude of actual XCO$_2$ plumes and thus supporting the application of PMIF to a large range of clumps.

Figure 1 shows that the uncertainty reduction on 6-hourly emissions from Paris before the satellite overpass can be up to 74% under calm wind condition (wind speed < 1 m s$^{-1}$) with the best observation sampling (in clear sky and with the satellite swath nearly centered on Paris), while it is systematically smaller than 45% for the 25th best observation sampling, over a full year of CS simulation. In addition, the uncertainty reductions have a large variation for narrow range of wind speeds, illustrating the strong impacts of the satellite track position with respect to the target and plume, together with the fraction of "clear sky" that modulates the sampling. In particular, the number of observations sampling the plume on the days when the wind direction is perpendicular to the satellite overpass tends to be less than the days when the wind direction is parallel to the satellite overpass. This is illustrated in Fig. 1 by the uncertainty reductions on the days when the wind speeds are 1.73 m s$^{-1}$, 7.6 m s$^{-1}$ and 8.1 m s$^{-1}$ that are lower than on the days with similar wind speeds.

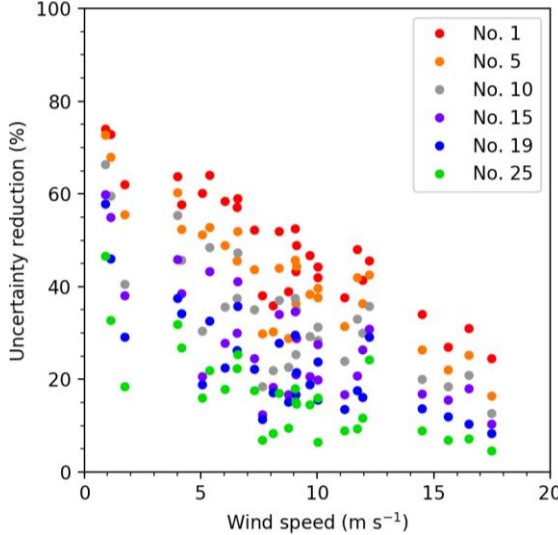

**Figure 1** Theoretical uncertainty reduction for the 6 h mean emissions in the PMIF-Paris experiments using the 1st (red), 5th (orange), 10th (light green), 15th (purple), 19th (blue) and 25th (green) best observation sampling from the CarbonSat simulation. The results from the 31 inversion days are given as a function of the average wind speed over the Paris clump. A comparison with the results from Broquet et al. (2018) is given in Fig. S3.

### 3.2 Potential of space observations for monitoring fossil fuel $CO_2$ emissions from individual clumps over 3 h time windows

Figure 2a shows the distribution of number of 8:30-11:30 time windows per clump for which the posterior uncertainty of 3 h mean emissions is smaller than 20% (this number is called N20) in Exp-NoCor. Clumps with small emission budgets tend to have lower N20 values than those with large budgets, due to the fact that the atmospheric plume generated by small emission clumps is difficult to distinguish from the measurement noise. Typically, N20 is smaller than 5 days for clumps emitting less than 2 MtC per year (like the city of Aswan, Egypt). Conversely, N20 is larger than 10 days for clumps emitting more than 2 MtC per year (like the cities of Manchester, UK, Boston, USA, and Chongqing, China). Note that clumps with emissions larger than 2 MtC, although representing less than 25% of the total number of clumps, contribute more than 83% of the total clump emissions. At regional scale (Figs. S4, S5), South America, North America, and Africa tend to have larger N20 values for same bin of clump annual emission than the other regions, while Middle East and Asia have the lowest ones. In addition, there are large variations and spatial heterogeneity in the N20 values within each emission bins (Fig. S5), which will be further discussed in Sect. 4.

We also show the numbers of 8:30-11:30 time windows per clump being labeled as "well-constrained" when the posterior uncertainty of 3 h mean emission is smaller than other thresholds, e.g. 10% and 30% (Fig. 2b). In general, using a posterior uncertainty larger than 20% as a threshold, we could expect more "well-constrained" cases. But for a given threshold, we still find the number of well-constrained cases increases with the emission budgets.

Figure 3 shows the posterior uncertainty in the clump emissions for the "well constrained" 8:30-11:30 time windows (identified in Exp-NoCor) from different OSSEs. It confirms that in all OSSEs, the posterior uncertainties for clumps with larger emissions are smaller than those with lower emissions. Within a given bin of clump annual emission, the posterior uncertainties from the various OSSEs are very similar, even though they are obtained with different hypothesis regarding the temporal auto-correlation in the prior uncertainty. The interpretation is that, for the inversion of the 3 h emissions before a

given satellite overpass, most of the constraint is imposed by the direct satellite observations during this overpass. These observations are independent on different days, and the constraints on different days are not strongly crossed even when errors in the prior estimate are highly correlated in time. However, although small, the impact of temporal auto-correlations in the prior uncertainties can be seen. For example, the posterior uncertainties in ASS (SCS) are systematically smaller than those in AMS (MCS), which confirms that the capability of the inversion system to use the information from observations from

previous/subsequent days to reduce the posterior uncertainties increases with the temporal auto-correlations. In SectCS, the posterior uncertainties are smaller than those in MCS and SCS in most regions (Fig. S5), due to the fact that the uncertainty in industrial emissions has a long temporal auto-correlation ($\tau_2$=180d).

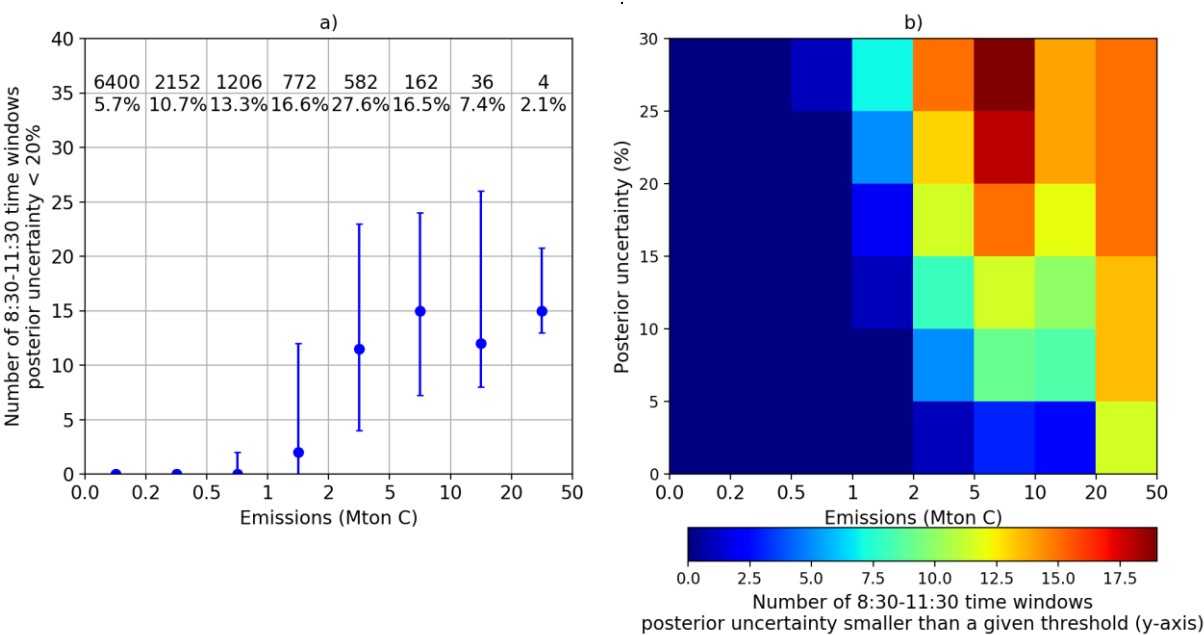

**Figure 2** a) Number of 8:30-11:30 time windows within a year for which the 3 h emissions are constrained with a posterior uncertainty less than 20% (N20) in the Exp-NoCor experiment. The results are binned according to clump annual emission with bin limits given on the x-axis of the figure. Dots and error bars are the median and interquartile range of N20 for all clumps within the emission bin. Numbers at the figure top indicate the number of clumps and the percentage of clump

emission within each bin. b) Number of 8:30-11:30 time windows (color) within a year for which the 3 h emissions are constrained with a posterior uncertainty less than a given threshold (y-axis) in the Exp-NoCor experiment.

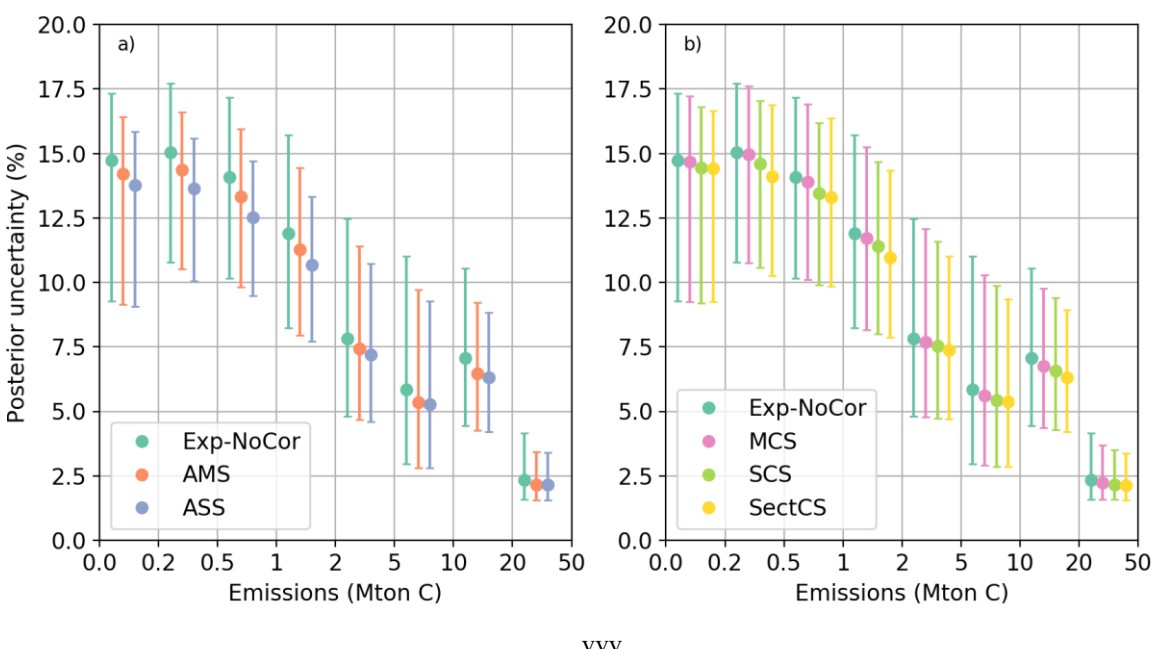

vvv

**Figure 3** Distribution of the posterior uncertainty in the 3 h mean emissions during the 8:30-11:30 time windows (for which the posterior uncertainty in 3 h mean emissions are smaller than 20% in Exp-NoCor) obtained with different OSSEs. Dots and error bars are the median and interquartile range. The results are binned according to the clump annual emission with bin limits given on the x-axis of the figure. Numbers at the figure top indicate the number of clumps and the percentage of clump emission within each bin.

### 3.3 Potential of space observations for monitoring daily fossil fuel CO$_2$ emissions

In previous sections, we analyzed the uncertainty reduction and the posterior uncertainty for the 3 h emissions that generate the atmospheric plume observed from space at 11:30. We now analyze the potential to monitor the daily emission, relying on the extrapolation of constraints on emissions between 8:30-11:30 using temporal auto-correlation of the prior uncertainties in the step 2 of the inversion (Sect. 2.7.2). Fig. 4 shows the distribution of the number of days when the posterior uncertainties in daily emissions are smaller than 20% (D20) for the same bins of emission clumps as in the previous section. Similar to the distribution of N20, clumps with small emission budgets tend to have lower D20 values than those with large budgets, due to having smaller signal-to-noise ratios for clumps with smaller emissions. The D20 values also strongly depend on the temporal auto-correlation in the prior uncertainty. When no correlation (Exp-NoCor) or short correlation (MCS) are assumed, D20 remains zero even for the largest clumps, since most of the daily emission are disconnected from the 3-hour

emissions that are constrained by the satellite observation and keep on bearing the large prior uncertainties associated with the Exp-NoCor and MCS scenarios. When significant temporal auto-correlations (e.g. in the case of AMS, ASS and SCS) are assumed, the results get better and the posterior uncertainties for the daily emissions become less than 20% for more than 100 days for clumps emitting more than 5 MtC per year. At regional scale (Fig. S6), the distribution of D20 values shows a similar pattern as N20: North America, South America and Africa have larger D20 values than Middle East and Asia for same bin of clump annual emission. But the distribution D20 values in SectCS have large regional variations, reflecting the regional differences in the share of emissions from different sectors.

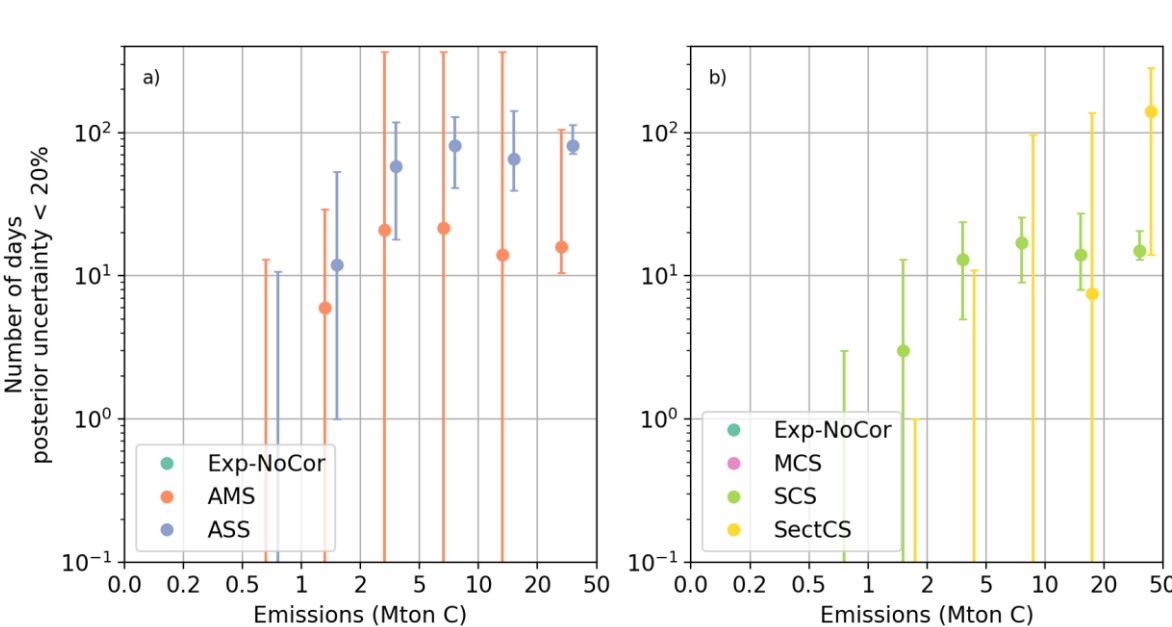

**Figure 4** Number of days within the year when the posterior uncertainty of daily emissions is smaller than 20% (D20). The results are binned according to the clump annual emission with bin limits given on the x-axis of the figure. Note that the median values of D20 for all clumps in Exp-NoCor and in MCS, for clumps whose annual emissions are between 0.5 MtC and 1 MtC in AMS, ASS and SCS, and for clumps whose emissions are below 10 MtC in SectCS, are all zero, so that the dots in these cases are not visible on y-axis with log scale. The dots and error bars are the median and interquartile range of D10 for all clumps within the emission bin. Numbers at the figure top indicate the number of clumps and the percentage of clump emission within each bin.

**3.4 Potential of space observations for monitoring annual fossil fuel CO$_2$ emissions**

We now analyze the results for the annual emissions, allowed again by the derivation of the posterior uncertainty covariance matrix **A** for individual clumps in step 2 of the inversion**,** and thus the aggregation of the posterior uncertainties in time. Figure 5 shows the posterior uncertainties in annual emissions from the OSSEs. When we assume that there is no temporal auto-correlations in the prior uncertainties, the uncertainties obtained from the inversions remain very close to the prior uncertainties (30%) for all emission bins since the information from the few well-constrained 8:30-11:30 time windows within

the year is not extrapolated to the huge unobserved fraction of the total annual emission over the year. The benefit of satellite observations becomes apparent when assuming that the prior uncertainties have temporal auto-correlations. Similar to the posterior uncertainties for 3 h emissions during 8:30-11:30, the posterior uncertainties in annual emissions are smaller in the OSSEs where the prior uncertainties have stronger temporal auto-correlation. This indicates that temporal auto-correlations help to extrapolate the information on the emissions from the satellite passes over a given clump to emissions during other

550    hours and days when there is no direct observations. Small clumps tend to have a larger relative posterior uncertainty in annual emissions than large clumps even when temporal error correlations are accounted for. The posterior uncertainties in the annual emissions of large cities with annual emission > 5 MtC per year can be constrained to better than 20% in AMS, SCS and SectCS, and to better than 10% in ASS. On the other hand, the posterior uncertainties for small emission clumps with annual emissions < 0.5 MtC per year are always larger than 15%, regardless of the temporal auto-correlations in prior uncertainties.

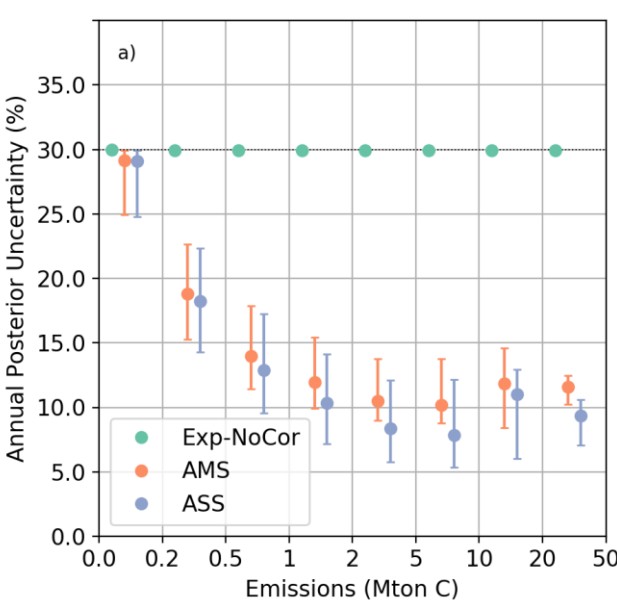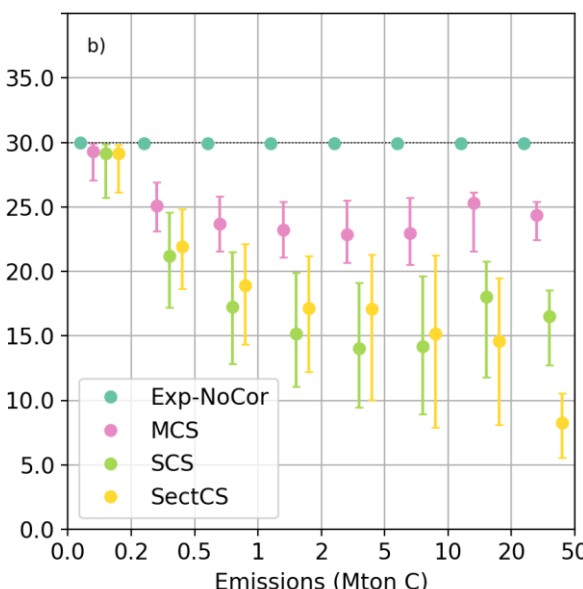

**Figure 5** Distribution of the posterior uncertainties in annual $CO_2$ emissions for different OSSEs. The results are binned according to the clump annual emission with bin limits given on the x-axis of the figure. Dots and error bars are the median and interquartile range of posterior uncertainty. Numbers at the figure top indicate the number of clumps and the percentage of clump emission within that bin.

## 4. Discussion and conclusions

PMIF provides information on the potential of space-borne imagery to constrain fossil fuel $CO_2$ emissions from emission clumps over the globe at the few-hour scale to the annual scale. It uses a simple Gaussian plume model to relate the emissions and the $XCO_2$ plumes. This is a strong simplification of the physics which impacts the range of uncertainties that can be accounted for in the inversion problem, but a preliminary evaluation against a more complex set-up (that of Broquet et al., 2018) indicates that it provides the correct order of magnitude for the uncertainties in the inverted emissions for an individual city: Paris.

In this study, we focused on the projection of uncertainties in satellite observations on the uncertainty of inverted emissions. Some sources of uncertainties that could have some impacts on the inversions when dealing with real data are ignored. Firstly, the plumes generated by the Gaussian plume model are straight along the wind direction at the source pixel. As a result, we allow the plumes from nearby clumps to potentially cross each other, but these plumes will systematically diverge on long distances. The Gaussian plume model cannot reproduce plumes overlapping along the atmospheric circulation like Eulerian transport models. In this sense, the overlapping effect of plumes can be underestimated in PMIF. In a realistic situation of atmospheric transport, if plumes from multiple clumps overlap very often, the inversion performance for individual clumps will be degraded since it will have the difficulties to accurately attribute the $XCO_2$ signals to individual clumps. Furthermore, we assume that the Gaussian plume model can perfectly link the emissions and $XCO_2$ and ignore the transport model error. If forced with erroneous wind fields, the simulation of $XCO_2$ plumes can have wrong shape and location, and thus generate large uncertainties in the inversions. In the inversion with actual $XCO_2$ observations from OCO-2, Nassar et al. (2017) allowed the wind direction to change from the wind re-analysis used to force the Gaussian plume model, if it improved the fit between simulated plumes and the observed signals. Reuter et al. (2019) and Kuhlmann et al. (2019) showed that the co-located $NO_2$ satellite observations could help to detect and constrain the location and shape of $XCO_2$ plumes. The transport model error may be partly reduced by incorporating additional information from other tracers when fitting the model to real data, but it is unknown to which extent these additional constraints is useful to improve the inversion of fossil fuel $CO_2$ emissions. With the current design of PMIF, the impact of transport error is hard to evaluate. Secondly, we ignore systematic measurement errors from the $XCO_2$ imagery. Broquet et al. (2018) showed that systematic error could hamper the ability of the inversion system to reduce the errors in the emissions estimates. Thirdly, we neglect the impact of uncertainties in diffuse fossil fuel $CO_2$ emissions (outside clumps) and non-fossil $CO_2$ fluxes (within and outside clumps), the latter including net ecosystem exchange (NEE) from the terrestrial biosphere, the $CO_2$ emitted by the burning of biofuel, the respiration from

human and animals (Ciais et al., 2020) and the net $CO_2$ fluxes between the atmosphere and ocean. For example, the signals from terrestrial NEE can be strong during the growing season, and the signals from ocean $CO_2$ fluxes may have a critical impact on the overall $XCO_2$ patterns in the proximity of coastlines. In principle, the signals of diffuse fossil fuel $CO_2$ emissions and non-fossil $CO_2$ fluxes outside the clumps can be potentially filtered by removing the local background $XCO_2$ field to extract plumes generated only by emissions from clumps (Kuhlmann et al., 2019; Reuter et al., 2019; Ye et al., 2020; Zheng et al., 2020). The non-fossil $CO_2$ fluxes within clumps vary from clump to clump, and could contribute a non-negligible fraction of the total $CO_2$ fluxes in many clumps (Bréon et al., 2015; Ciais et al., 2020; Wu et al., 2018a). The satellite observations alone cannot effectively differentiate the fossil fuel $CO_2$ emissions and the non-fossil $CO_2$ fluxes within clumps. In the clumps with non-negligible non-fossil $CO_2$ fluxes, the inversion of fossil fuel $CO_2$ emissions could be influenced (Ye et al., 2020; Yin et al., 2019). Fourthly, the PMIF system controls the scaling factors for the mean emissions of daily 3-h and 21-h windows and for each clump, ignoring uncertainties in the spatial distribution and temporal profile of the emissions (described by the operator $\mathbf{M}_{inventory}$) within the clumps and over the time windows. Such uncertainties are called aggregation errors (Wang et al., 2017; Wu et al., 2011). However, Broquet et al. (2018) compared the results of inversions using the realistic spatial distribution of emissions and using a homogenous one over two discs with different radius for $\mathbf{M}_{inventory}$, and found that having imperfect spatial distribution of emissions to model $\mathbf{M}_{inventory}$ (thus the aggregation error) only has a small impact on the uncertainties and errors in the inverted emissions. Future developments in PMIF should attempt at quantifying the impacts of such sources of uncertainties, while keeping its power of constraining the emissions from a large range of sources with global coverage.

Although it ignores the sources of uncertainties listed above, the current PMIF can still be used to investigate the impacts of some key parameters of inversion problem and to allow, for the first time, to make a first-order extrapolation of the results from single-city studies to all significant emission clumps over the globe and under a full range of meteorological conditions during a year.

The key result summarized in Figure 2 is that using a single CO2M satellite, only the clumps with annual budget higher than 2 MtC per year (e.g. Manchester, UK, Boston, USA and Chongqing, China) can potentially be well constrained with N20 being larger than 10 within a year. However, there are large variations in the N20 values for clumps with such levels of emission. Figures 6a and 6b show the maps of the number of observations within each 2º×2º grid cell during one year in the USA and China, which is an indicator for the frequency of clear-sky days: the larger the number of observations, the higher frequency of clear-sky days. It is clearly seen in Fig. 6c and 6d that the clumps in Southern China have low N20 values when they are located in areas with a low frequency of clear-sky days. For clumps that have emissions between 2 and 5 MtC per year, N20 values are below 10 days in a cloudy/hazy region like Southeastern China, and are close to 30 days in a clear-sky region like the Western Coast of the USA. These results illustrate the dependence of the potential of satellite observations to constrain emissions on the frequency of clear-sky conditions. The relative uncertainty in the inversion of the emissions from a clump is primarily driven by the budget of these emissions, and by the wind speed (as illustrated by Fig. 1). The frequency of clear-sky days modulates the number of direct observation of the plume from a clump and thus the number of days for which the

625 inversion can decrease the uncertainty when ignoring temporal auto-correlations in the prior uncertainty in Exp-NoCor. The frequency of clear-sky day, together with the emission rate and wind speed, are the main drivers of the posterior uncertainty in daily to annual emissions when accounting for temporal auto-correlations in the prior uncertainty.

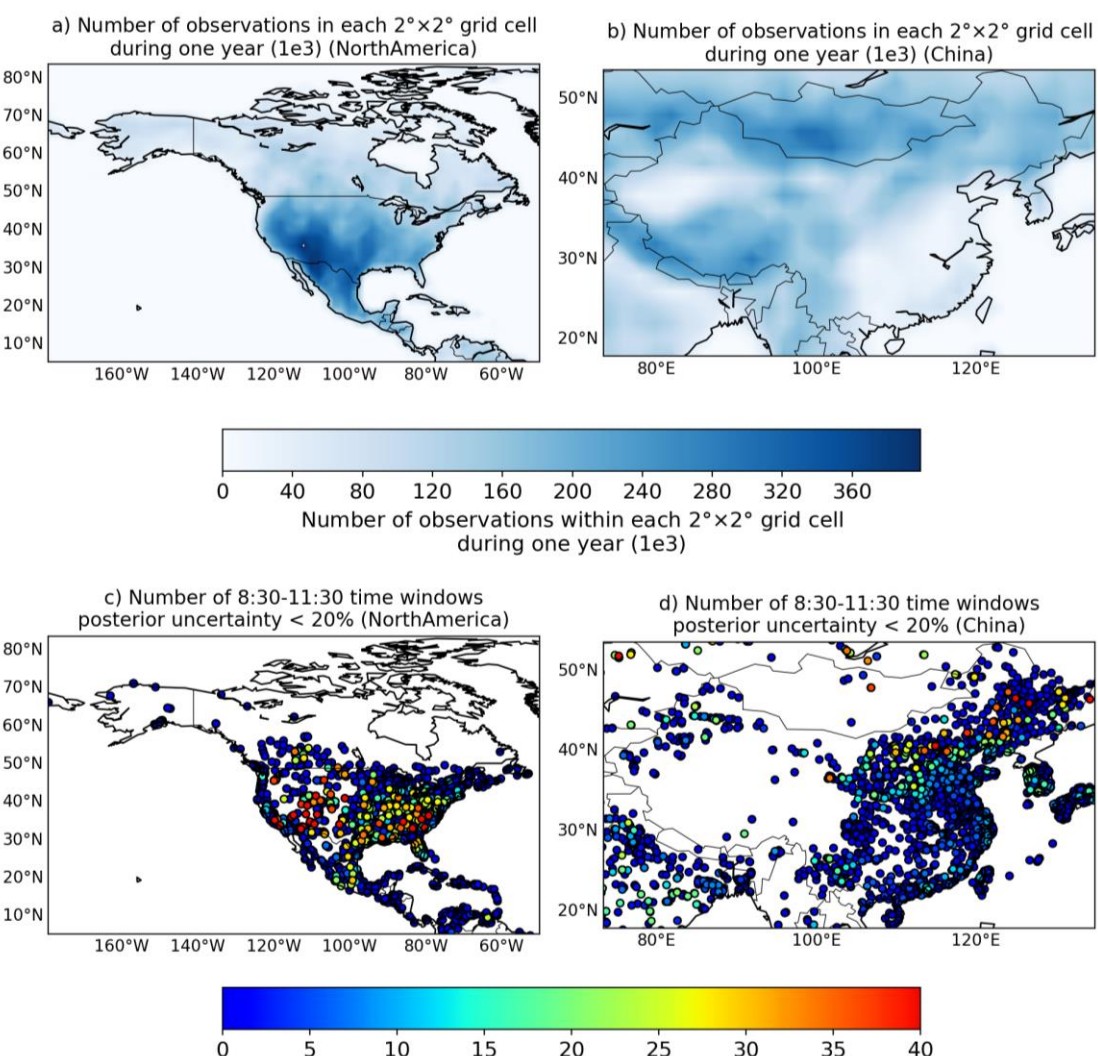

**Figure 6** Number of observations in 2°×2° grid cells during one year (a and b) and N20 values (c and d).

We showed that one CO2M imager can provide a direct constraint for the estimate of emissions from clumps with emissions larger than 2 MtC per year, but over limited periods only. N20 is smaller than 25 for most clumps, indicating that even for emissions during 8:30-11:30, one cannot expect more than 25 days when the CO2M observations sample the plumes
from clumps with sufficient number of observations (Fig. 2) during one year. The use of a constellation of CO2M satellites in

the current plan could potentially improve the frequency of good samplings. Imaging from geostationary orbit (GEO) imagers like NASA's GeoCarb mission (O'Brien et al., 2016; Polonsky et al., 2014) could offer sampling during different periods within a day to constrain the diurnal profile of emissions. Highly elliptical orbit (HEO) imagers could also provide observations at northern high latitudes with a similar high frequency as GEO (Nassar et al., 2014). However, even though multiple space-borne platforms can sample the plumes more frequently, the satellites using passive sensors like that planed for CO2M can never sample the plumes on cloudy/hazy conditions.

We also investigated the possibility of extrapolating the information obtained from the time windows for which the emissions are constrained by satellite observations to estimate emissions on other hours, days and through a year. Such an extrapolation relies on the model of the emission inventories used as a prior of PMIF, that is, in the framework of PMIF, the temporal auto-correlation of the uncertainty of prior emissions. The analysis of posterior uncertainties in the 3 h mean emissions, in daily emissions and in annual emissions all show that the configuration of this temporal auto-correlation has a large impact on the inversion results. For example, posterior uncertainties in annual emissions range from less than 10% with strong auto-correlation (ASS) to 25% with medium auto-correlation (MCS) for clumps with emissions higher than 2 MtC per year. The orders of magnitude in the posterior uncertainty will be critical to the objective assessment of annual emissions. However, since state-of-the-art emission products rarely report their uncertainties and temporal auto-correlations (Andres et al., 2016; Gurney et al., 2019), it is difficult to exclude any configuration of OSSEs in this study. The strong impact of the prior uncertainty on the inversion results thus highlights the priority of future researches to systematically assess the uncertainty, especially the temporal error co-variances, in the emission products.

Even if emissions can be effectively constrained by CO2M for clumps whose emissions are larger than 2 MtC per year, the sum of annual emission budgets from these large clumps account only for 54% of the total $CO_2$ clump emissions and for 36% of the total global fossil fuel $CO_2$ emissions (accounting for diffuse emissions outside the clumps), according to the clump definition of Wang et al. (2019) and the ODIAC emission map. For a specific country, clumps with emissions larger than 2 MtC per year typically represent less than 50% of the total national emissions (accounting for diffuse emissions outside the clumps). It thus shows the difficulty to use a single CO2M imager as the only source of information to constrain national emissions. This limitation of a single CO2M imager calls for innovations to integrate other types of observations in inversion systems to improve the ability to estimate emissions at both city scale (Lauvaux et al., 2016; Sargent et al., 2018; Staufer et al., 2016) and larger spatial scales (Palmer et al., 2018; Wang et al., 2018).

## 5. Code availability

The source code for PMIFv1.0 is included in the Supplement. To run PMIF, some input files are needed. The ODIAC inventory is available at http://db.cger.nies.go.jp/dataset/ODIAC/DL_odiac2018.html. The clump dataset is available at https://doi.org/10.6084/m9.figshare.7217726.v1. The list of clump information (e.g. index, latitude and longitude of the center),

which is also needed as an input, is included in the Supplement. The wind fields from CCMP are available at http://www.remss.com/measurements/ccmp/. EDGAR v4.3.2 emission maps are needed to run the SectCS inversion, and are available at https://edgar.jrc.ec.europa.eu/overview.php?v=432_GHG.

**Author contributions**

PC, GB and FMB designed the research; YW and FL developed the PMIF code and made the analysis; MB and MR simulated of satellite sampling and random measurement noise for CarbonSat and CO2M imagers; YW, GB, FMB, FL, MB, MR, YM, AL, GLM, BZ and PC wrote the paper.

**Acknowledgement**

This work was mainly conducted and funded in the frame of the ESA projectNo.4000120184/17/NL/FF/mg. It also received support from the TRACE Industrial Chair (UVSQ / CEA / CNRS / Thales Alenia Space / TOTAL / SUEZ) ANR-17-CHIN-0004 funded by the program « Chaires Industrielles 2017 » of ANR. We would like to thank Bernard Pinty for providing the vision of a CO2 Monitoring and Verification Support (MVS) Capacity within the framework of the EU's Copernicus Programme.

**Appendix: Acronyms**

AMS: Annual component and Moderately correlated Sub-annual component

ASS: Annual component and Strongly correlated Sub-annual component

CDIAC: Carbon Dioxide Information Analysis Center

CNES: Centre National d'Etudes Spatiales

CO2M: Copernicus Anthropogenic Carbon Dioxide Monitoring

D20: Number of days within the year when the posterior uncertainty of daily emissions is smaller than 20%

ECMWF: European Centre for Medium-Range Weather Forecasts

ESA: European Space Agency

EUMETSAT: European Organisation for the Exploitation of Meteorological Satellites

GOSAT: Greenhouse Gases Observing Satellite

MCS: Moderately Correlated Sub-annual component

N20: number of 8:30-11:30 time windows per clump for which the posterior uncertainty of 3 h mean emissions is smaller than 20%

NoCor: No temporal auto-correlation

OCO: Orbiting Carbon Observatory

ODIAC: Open-source Data Inventory for Anthropogenic $CO_2$

OSSE: Observing System Simulation Experiment

PMIF: Plume Monitoring Inversion Framework

SCS: Strongly Correlated Sub-annual component

SectCS: Sector-dependent Correlated Sub-annual component

SZA: solar zenith angle

TIMES: Temporal Improvements for Modeling Emissions by Scaling

$XCO_2$: vertically integrated columns of dry-air mole fractions of $CO_2$

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
