# Peer review of "PMIF v1.0: an inversion system to estimate the potential of satellite observations to monitor fossil fuel CO2 emissions"

_Geoscientific Model Development, 2019_

## Author Comment (AC1) · 23 Jan 2020

We would like to revise the Acknowledgement section to highlight the support and funding from ESA contract. The revised texts are as follows:

This work was mainly conducted and funded in the frame of the ESA project No.4000120184/17/NL/FF/mg. It also received support and funding from the TRACE Industrial Chair (UVSQ / CEA / CNRS / Thales Alenia Space / TOTAL / SUEZ) ANR-17-CHIN-0004 funded by the program "Chaires Industrielles 2017" of ANR. We would like to thank Bernard Pinty for providing the vision of a CO2 Monitoring and Verification Support (MVS) Capacity within the framework of the EU's Copernicus Programme.

[Figure]
[Figure]

---

## Referee Comment (RC1) · Anonymous Referee #1 · 18 Feb 2020

**1   Overview:**

Review of "*PMIF v1.0: an inversion system to estimate the potential of satellite observations to monitor fossil fuel CO2 emissions*" by Wang *et al.*

Wang *et al.* present an OSSE framework to estimate error reductions for a proposed satellite. It's based on a Gaussian plume that they run for many emission hotspots. They've done this over a large domain (globally) at fairly high spatial resolution (2 km). The work is interesting but the description of the methods could use quite a bit of work. There are some important steps in the actual implementation that are quite convoluted.

Fixing this seems like a critical for publication in a journal focused on geoscientific model development. I suggest major revisions for the manuscript.

**2 Comments:**

**2.1 Solution to their inversion**

I'd prefer the authors not use $\mathbf{A}$ as the posterior covariance matrix, I usually think of $\mathbf{A}$ as the averaging kernel. This is particularly confusing because you are solving for emission reductions that are the diagonals of the averaging kernel matrix. In any case, Supplemental Section 1 presents what the authors are *actually* doing, which differs from the equations they present in Eq. 1 and 2. In Supplemental Section 1 the authors present a derivation that is both important and convoluted. It's unclear if this is something the authors devised themselves or if it follows from other work. Typically when people decompose error covariance matrices into spatial and temporal components they use a Kronecker product (e.g., Yadav & Michalak, GMD 2013). The Kronecker product greatly reduces the computational expense. The assumptions that go into a Kronecker product are also easy to follow because it is widely used. It's also amenable to sparse matrices (I'm assuming the authors are using sparse matrices).

I think the authors should remove Equation 2 and bring Supplemental Section 1 into the main text. Supplemental Section 1 is important because this is what they are *actually* doing. This seems like the main contribution to me.

Finally, I would strongly suggest not using "pseudo" in Supplemental Section 1 because that implies computing a pseudoinverse, which has a very specific mathematical definition. Unless, of course, the authors are computing a pseudoinverse in which case that should be made clear.

The authors should change the title. It's not an *inversion framework* as they are not estimating fluxes.

**2.2 Justification on the use of a Gaussian plume**

Real plumes are only Gaussian in the time-averaged sense. The satellite observations provide a snapshot in time that likely would not be Gaussian. I think the authors need to provide some justification as to why a Gaussian plume is appropriate for data that is not time-averaged. A Gaussian plume may give a reasonable upper bound on the uncertainty reduction, but will likely induce systematic biases if implemented operationally. These potential biases should be discussed.

The authors should give more explanation of $\sigma_j$. There are two parameters in a Gaussian plume model and they spend one line talking about $\sigma_j$: "The $\sigma_j$ is a function of downwind distance $i$ and atmospheric stability parameter. We take the form for $\sigma_j$ from Ars et al. (2017).".

**2.3 Clumps**

I don't like the terminology "emission clumps". It doesn't fit with the actual definition of a clump:
noun: *"a compacted mass or lump of something"*
verb: *"form into a clump or mass"*

Emissions don't clump. The various sources have just been grouped together. The abstract of their 2019 paper seemed to use "hotspot" and "clusters" which I would prefer to "clump". A cluster would be a much more intuitive name for this.

**2.4   References**

The authors show a very strong bias towards European studies. They don't seem to mention any of Ray Nasser's work in the intro even though his 2017 GRL paper used a Gaussian plume model with satellite observations to study individual sources. They also seem to have missed Eric Kort's work using GOSAT to study megacities (Kort et al., GRL 2012; among others).

**2.5   3 hours vs 6 hours**

Why is there a 6-hour window for Paris and a 3-hour window globally? I see, it's defined afterward. This should be moved forward to explain why Broquet chose 6 hours and why they choose 3 hours. How is 3 hours chosen? It seems to just be picked randomly.

**3   Specific comments:**

Title: Remove fossil fuel from the title. I don't see how they could differentiate fossil from non-fossil sources in their analysis.

Section 2.1: Should reference the sections that define the error covariance parameters.

Line 126: what is $y^{\text{fixed}}$?

Line 181: rephrase, too colloquial: "but the PMIF can hardly handle hourly emissions when covering a whole year".

---

## Referee Comment (RC2) · Anonymous Referee #2 · 24 Feb 2020

This study assesses the potential of satellite imagery of a future mission CO2M XCO$_2$ to constrain the emissions from cities and power plants over the whole globe for one year. To reduce the computational cost of the traditionally used 3-D full transport models, this study simplified the observation operator with a few idealized hypotheses: (a) a Gaussian plume model, no model errors, (b) no overlapping effects from nearby hotspots, (c) no impact of natural carbon cycle fluxes. It is useful to get a global-scale estimate for the potential of emission uncertainty reductions for the proposed mission – even though the results are not very positive in terms of CO$_2$ measurements' potential in constraining fossil fuel CO$_2$ emissions alone given those idealized setups.

General comments:

The authors highlight the global scope of this study, but no global distribution is shown. Fig. 6 shows information about US and China, why only these two regions? The global results are aggregated with emission density bins (Fig 2 - 5), which I assume is not the only determining factor. With simple statistics of median spread, a lot of information is lost. It does not really provide a "global" view. Fig. 1 highlighted the impacts of wind speed, which may create spatial patterns that overlay with emission density maps. Such information may reveal a better global overview.

Also, a posterior uncertainty of 20% has been used as a benchmark throughout the paper (given a 30% prior uncertainty). However, only a few cases/days can meet such a requirement. Thus, it may be more helpful to show what posterior uncertainty can be achieved for a given length of days across typical regions (e.g., using a 2-D matrix?)

A few technical points:

-L35: "more than 10 times within one year" is a low number. As stated above, if this is the case, is using 20% as the only threshold discussed in the paper a reasonable choice?

-L58-59: other studies worth mentioning, for instance:

Kort, E. A., Frankenberg, C., Miller, C. E. and Oda, T.: Space-based observations of megacity carbon dioxide, Geophys. Res. Lett., 39(17), n/a-n/a, doi:10.1029/2012GL052738, 2012.

Nassar, R., Hill, T. G., McLinden, C. A., Wunch, D., Jones, D. B. A. and Crisp, D.: Quantifying $CO_2$ Emissions From Individual Power Plants From Space, Geophys. Res. Lett., 44(19), 10,045-10,053, doi:10.1002/2017GL074702, 2017.

Schwandner, F. M., Gunson, M. R., Miller, C. E., Carn, S. A., Eldering, A., Krings, T., Verhulst, K. R., Schimel, D. S., Nguyen, H. M., Crisp, D., O'Dell, C. W., Osterman, G. B., Iraci, L. T. and Podolske, J. R.: Spaceborne detection of localized carbon dioxide

sources., Science, 358(6360), eaam5782, doi:10.1126/science.aam5782, 2017.

-L102: "for the first time" - It is important to talk about the bright side, however, it is equally important to define the underlying assumptions clearly. The discussion came later, but I believe a higher level of clarification here will be helpful.

-L105: How about observations near the edge of the swath? The resolution would change accordingly.

-L137: $y_{fixed}$ is not explained.

-L144, 148: "In this study" is used quite a lot. Not all necessary.

-L152: not accounting for diffuse $CO_2$ fluxes is an important distinction. It is an important assumption that needs to be emphasized as the natural carbon cycle will have a strong imprint in many areas.

-L225: a simple description of the sigma parameter (e.g., what determines it) will help the reader without having to refer to Ars et al. (2017).

-L369: why not just use Fig. S3 for side by side comparison?

-L404: "N20". There are quite some acronyms already that need checking back and forth. Will improve the reading removing some that do not have intuitive meanings.

-L501: How about the optimized state? Curious how well will the Gaussian Plum model do if it assimilates the psuedo observations generated using the full 3-D models in this case. It will be a strong demonstration if it can get the emission order general variations right!

-L519: Quite a few studies explore the interfering effect of natural $CO_2$ fluxes.

Wu, K., Lauvaux, T., Davis, K. J., Deng, A., Lopez Coto, I., Gurney, K. R. and Pataraska, R.: Joint inverse estimation of fossil fuel and biogenic CO2 fluxes in an urban environment: An observing system simulation experiment to assess the impact of multiple uncertainties, Elem Sci Anth, 6(1), 17, doi:10.1525/elementa.138, 2018.

Yin, Y., Bowman, K., Bloom, A., Worden, J.: Detection of fossil fuel emission trends in the presence of natural carbon cycle variability, Environmental Research Letter, 14(8):084050, doi:10.1088/1748- 9326/ab2dd7, 2019.

-L538: Again, I understand that 20% posterior uncertainty is a desirable goal, but it did not provide a full picture if the values for the high emission densities are only at the order of 10 days for a year. Other references will help define the landscape.

-Figure 3: the number of clamps is repeated in every plot from Fig. 3-5. Reductant to repeat so many times. Maybe indicate clearly that (a) and (b) are the same just for different experiments.

―――――――――――――――――――

---

## Author Comment (AC3) · 24 Jul 2020

**Response to comments on "PMIF v1.0: an inversion system to estimate the potential of satellite observations to monitor fossil fuel CO2 emissions" by Y. Wang et al.**

We thank the referee for reviewing our manuscript. Please find attached a point-by point reply (in black) to each of the comments raised by the referee (in blue) with legible text and figures organized along the text. For your convenience, changes in the revised manuscript are highlighted with dark red. All the pages and line numbers correspond to the original version of text.

This study assesses the potential of satellite imagery of a future mission CO2M XCO2 to constrain the emissions from cities and power plants over the whole globe for one year. To reduce the computational cost of the traditionally used 3-D full transport models, this study simplified the observation operator with a few idealized hypotheses: (a) a Gaussian plume model, no model errors, (b) no overlapping effects from nearby hotspots, (c) no impact of natural carbon cycle fluxes. It is useful to get a global-scale estimate for the potential of emission uncertainty reductions for the proposed mission – even though the results are not very positive in terms of  $CO_2$  measurements' potential in constraining fossil fuel  $CO_2$  emissions alone given those idealized setups.

**Response:**

We would like to clarify the point (b) listed above by the reviewer. Actually, there can be some overlapping between the plumes generated by nearby clumps in the PMIF. In Eulerian transport model, the plumes from nearby sources can converge along atmospheric circulation. However, here, since using a classical Gaussian plume model, the plumes are straight along the wind direction. Therefore, the plumes from two nearby clumps can cross each other, but they'll systematically diverge on long distances, which, in some cases, can lead to a significant underestimation of the plume overlapping. To make it clearer, we revised the sentences Ln 506-508: "...Firstly, the plumes generated by the Gaussian plume model are straight along the wind direction at the source pixel. As a result, we allow the plumes from nearby clumps to potentially cross each other, but these plumes overlapping along the atmospheric circulation like Eulerian transport models. In this sense, the overlapping effect of plumes can be underestimated in PMIF. In a realistic situation of atmospheric transport, if plumes from multiple clumps overlap very often, the inversion performance for individual clumps will be degraded since it will have the difficulties to accurately attribute the XCO2 signals to individual clumps."

**General comments:**

The authors highlight the global scope of this study, but no global distribution is shown. Fig. 6 shows information about US and China, why only these two regions? The global results are aggregated with emission density bins (Fig 2 - 5), which I assume is not the only determining factor. With simple statistics of median spread, a lot of information is lost. It does not really provide a "global" view. Fig. 1 highlighted the impacts of wind speed, which may create spatial patterns that overlay with emission density maps. Such information may reveal a better global overview.

**Response:**

We synthesize the global results with the plot of median values and the spread in Figs. 2-5. Figure 6 is shown to prove that the frequency of clear-sky largely explains the large variations within each emission bin. We agree with the reviewer that the inversion results are mainly driven by a combination of emission rates, wind speed and frequency of clear-sky days. However, plotting clumps' uncertainty on top of clump emissions or wind speed would make the figure too saturated to read. (Figure 6c and d are already close to a saturation of dots). Following the reviewer advice, we have produced figures like Figure 6c and d for all the regions of the globe. However, since they do not bring much more qualitative insights than Figure 6c and d, we have put them in the supplementary material. In the main text, we remind the readers to refer to these figures accordingly:

Ln 410: "At regional scale (Figs. S4, S5), South America, North America, and Africa tend to have larger N20 values for same bin of clump annual emission than the other regions, while Middle East and Asia have the lowest ones. In addition, there are large variations and spatial heterogeneity in the N20 values within each emission bins (Fig. S5), which will be further discussed in Sect. 4."

Ln 545: "... These results illustrate the dependence of the potential of satellite observations to constrain emissions on the frequency of clear-sky conditions. The relative uncertainty in the inversion of the emissions from a clump is primarily driven by the budget of these emissions, and by the wind speed (as illustrated by Fig. 1). The frequency of clear-sky days modulates the number of direct observation of the plume from a clump and thus the number of days for which the inversion can decrease the uncertainty when ignoring temporal auto-correlations in the prior uncertainty in Exp-NoCor. The frequency of clear-sky day, together with the emission rate and wind speed, are the main drivers of the posterior uncertainty in daily to annual emissions when accounting for temporal auto-correlations in the prior uncertainty."

Also, a posterior uncertainty of 20% has been used as a benchmark throughout the paper (given a 30% prior uncertainty). However, only a few cases/days can meet such a requirement. Thus, it may be more helpful to show what posterior uncertainty can be achieved for a given length of days across typical regions (e.g., using a 2-D matrix?)

**Response:**

Firstly, we stress that the prior uncertainties are different at different time scale. In all the experiments, the prior uncertainty is 30% for annual emissions. When decomposing the uncertainty of annual emissions to the scales of 3 h and 21 h time windows, the resulting uncertainties largely depend on the assumption about the temporal auto-correlations (Sect. 2.6). In the ASS scenario, the prior uncertainty for 3 h emissions is  $\sqrt{(44\%^2+26\%^2)}=51\%$ , while in NoCor scenario, it is 614%.

Eq. (1) shows that the posterior uncertainty and uncertainty reduction depend on the prior uncertainty. For example, if the projection of uncertainties in satellite observations on the uncertainty in emissions (i.e.  $\mathbf{M}^{T}\mathbf{R}^{-1}\mathbf{M}$ ) equals to 50% for a single 3 h time window, in ASS scenario, the posterior uncertainty equals to  $\sqrt{1/(1/(51\%)^{2}+1/(50\%)^{2})}=36\%$ , while in NoCor, the posterior uncertainty equals to 50%. In this situation, if the benchmark is chosen too high (e.g. 50%), it is too easy for ASS scenario, while it still requires a lot of constraints from satellite observations in NoCor scenario. If we choose 60% as the benchmark for assessing the posterior uncertainty, then the prior uncertainty in emissions in ASS will always below the

benchmark, even without conducting the inversion. Given different values of prior uncertainty in different scenarios, it is not easy to find a metric to fairly compare the results from different scenarios. We choose 20% as a benchmark because if the posterior uncertainty is below 20%, it is mainly determined by the projection of uncertainties in satellite observations on the uncertainty of emissions.

Furthermore, the posterior uncertainty in the emissions within 3 h time window or in the daily emissions, and thus the number of N20 and D20 are among the diagnostics we investigated on the potential of satellite observations. We also assessed the posterior uncertainty at annual scale, which integrates the uncertainty in all time windows, not only those whose uncertainty is smaller than 20%.

In the first version of this paper, we did consider to use a 2-D matrix to show the results, as shown in Fig. R2. We think such a 2-D matrix plot has its own disadvantages: 1) as stated above, the posterior uncertainty also depends on the prior uncertainty, if the threshold is chosen high, it does not properly represent the actual constraints from satellite observations; 2) such a plot cannot show the large variations in the number of cases within each emission bin. But this information is easy to read from the whisker plot in Fig. 3-5; and 3) such a 2-D matrix plot cannot compare the performance of the inversion in different experiments directly. Given the close values of some experiments (e.g. AMS and ASS in Fig. 3), the difference between experiments cannot be noticed by eye from separate 2-D matrix plots. Given these considerations, we decided to use the plots that have been shown in the paper, which can synthesize as the most information as we want to deliver, and also makes it possible to compare the performance for different experiments.

**Figure R2** Number of 8:30-11:30 time windows (color) within a year for which the 3 h emissions are constrained with a posterior uncertainty less than a given threshold (y-axis) in the Exp-NoCor experiment.

In the revised manuscript, we add in Fig. 2 the 2-D matrix plot to illustrate the number of cases under different threshold. But we do not do that for the other diagnostics. And we add some discussions about this figure:

"At regional scale (Fig. S4), South America, North America, and Africa tend to have larger N20 values for same bin of clump annual emission than the other regions, while Middle East and Asia have the lowest ones. In addition, there are large variations and spatial heterogeneity in the N20 values within each emission bins (Fig. S5), which will be further discussed in Sect. 4.

We also show the numbers of 8:30-11:30 time windows per clump being labeled as "well-constrained" when the posterior uncertainty of 3 h mean emission is smaller than other thresholds, e.g. 10% and 30% (Fig. 2b). In general, using a posterior uncertainty larger than 20% as a threshold, we could expect more "well-constrained" cases. But for a given threshold, we still find the number of well-constrained cases increases with the emission budgets."

**A few technical points:**

-L35: "more than 10 times within one year" is a low number. As stated above, if this is the case, is using 20% as the only threshold discussed in the paper a reasonable choice?

**Response:**

See our discussion above about the choice of N20 as the main diagnostic to characterize the frequency of "well constrained" inversions.

**-L58-59: other studies worth mentioning, for instance:**

Kort, E. A., Frankenberg, C., Miller, C. E. and Oda, T.: Space-based observations of megacity carbon dioxide, Geophys. Res. Lett., 39(17), n/a-n/a, doi:10.1029/2012GL052738, 2012.

Nassar, R., Hill, T. G., McLinden, C. A., Wunch, D., Jones, D. B. A. and Crisp, D.: Quantifying CO2 Emissions From Individual Power Plants From Space, Geophys. Res. Lett., 44(19), 10,045-10,053, doi:10.1002/2017GL074702, 2017.

Schwandner, F. M., Gunson, M. R., Miller, C. E., Carn, S. A., Eldering, A., Krings, T., Verhulst, K. R., Schimel, D. S., Nguyen, H. M., Crisp, D., O'Dell, C. W., Osterman, G. B., Iraci, L. T. and Podolske, J. R.: Spaceborne detection of localized carbon dioxide sources., Science, 358(6360), eaam5782, doi:10.1126/science.aam5782, 2017.

**Response:**

Thanks for the reviewer to remind some more references. In the revised introduction, we rewrite the paragraph:

Ln 55 "... Alternatively, vertically integrated columns of dry-air mole fractions of  $CO_2$  (XCO2) from satellites offer the opportunity to sample the atmosphere with a global coverage. Kort et al. (2012) and Janardanan (2016) found that significant XCO2 enhancements could be detected over some megacities using Greenhouse Gases Observing Satellite (GOSAT) XCO2 observations. Schwandner et al. (2017) also found XCO2 enhancements of 4.4 to 6.1 ppm in the Los Angeles urban CO2 dome using observations from Orbiting Carbon Observatory-2 (OCO-2). Nassar et al. (2017) used the XCO2 observations from OCO-2 to quantify CO2 emissions from several middle- to large-sized coal power plants. However, the design of GOSAT and OCO-2 observations with sparse sampling was focused on the monitoring of CO2 natural fluxes. Recent studies show a limited amount of clear detections of transects of XCO2 plumes from cities or plants in OCO-2 observations (Zheng et al., 2020a) so that GOSAT and OCO-2 data keep on being hardly used to estimate CO2 city emissions. The potential for reducing uncertainties in fossil fuel CO2 emissions at the scale of point sources (Bovensmann et al., 2010), cities (Broquet et al., 2018; Pillai et al., 2016) and agglomerations of several cities (O'Brien et al., 2016) should dramatically change with the planned satellite missions with

**imaging capabilities. These studies consistently showed that ....."**

-L102: "for the first time" - It is important to talk about the bright side, however, it is equally important to define the underlying assumptions clearly. The discussion came later, but I believe a higher level of clarification here will be helpful.

**Response:**

We revise the sentences Ln 101-105: "Therefore, in this study, we develop a Plume Monitoring Inversion Framework (PMIF) and conduct a set of Observing System Simulation Experiments (OSSEs) to assess, for the first time, the performance of a satellite instrument to monitor the emissions of all the clumps across the globe and over a whole year. The imager studied has the foreseen characteristics of the individual satellites of the forthcoming CO2M mission. It would be a high-resolution spectrometer, with  $2 \text{ km} \times 2 \text{ km}$  resolution pixels and a swath of 300 km, and it would be placed on a sun-synchronous orbit ensuring global coverage in 4 days. The PMIF inversion system relies on the list of clumps extracted by Wang et al. (2019) from the ODIAC inventory, on the Gaussian plume model to simulate the XCO2 plumes generated by the emissions from these clumps, on an analytical inverse modeling framework, and on a combination of overlapping assimilation windows to solve for the inversion problem over the globe and a full year. It also addresses the question of temporal extrapolation that is needed to generate estimates of annual emissions from the information of a limited number of time windows for which emissions are well constrained by the direct satellite images, by accounting for the temporal auto-correlation of the prior uncertainties. The performance is assessed in terms of the uncertainties in the emissions (Sect. 2.1) at different scales. The PMIF uses a Gaussian plume model at the local scale to ensure that the computation cost is affordable. Such a model can often hardly fit with actual plumes over the distances considered in this study (due to variations in the wind field, topography, vertical mixing etc. over such distances) but is shown, when driven with suitable parameters, to provide a satisfactory simulation of the plume extent and amplitudes, which appear to be the main drivers of the targeted computations of uncertainties in the emissions in our OSSE framework (as shown in section 3.1). In PMIF, we also ignore the impact of some sources of uncertainties on the inversion of emissions, including systematic errors on the XCO2 retrievals, the impact of uncertainties in diffuse anthropogenic emissions outside clumps, in natural CO2 fluxes (within and outside clumps), and in the spatial and temporal variations of emissions within the clump and the short time windows that the inversion aims to solve. These impacts are discussed in detail afterwards."

**-L105: How about observations near the edge of the swath? The resolution would change accordingly.**

**Response:**

The observations are simulated using the method and model described by Buchwitz et al. (2013) in the frame of different ESA projects studying XCO2 imagers with inputs from ESA. Different values for the parameters in the model are used to account for the differences between the original configuration for CarbonSat and the configuration for CO2M.

The edge effect is small because the swath width we discussed is only 300 km. For a satellite at 700 km altitude and with a ground pixel at nadir at the resolution of 2 km, the resolution of a pixel at the edge of the swath is about 2.09 km, which is still very close to 2 km.

In fact, the edge effect is very small and very well within the overall uncertainty of the method which is based on various input data sets.

**-L137: yfixed is not explained.**

**Response:**

We revised the sentence:

"... The inversion derives a statistical estimate for a set of control variables x in a model  $x \rightarrow y=Mx$  that simulates the satellite XCO2 measurements  $y^{\circ}$ . The model **M** linking x and y is a combination of flux and atmospheric transport models (detailed in Sect. 2.4), and is called observation operator hereafter. As explained below, we do not have a constant term added to **M**x in the observation operator of the PMIF that would gather the atmospheric CO2 signature of the fluxes not controlled by the inversion (like non-fossil fluxes and the background XCO2 field) since the uncertainty in such fluxes is ignored. The inversion follows a Bayesian statistical framework,..."

-L144, 148: "In this study" is used quite a lot. Not all necessary.

**Response:**

We have gone through the manuscript carefully, and removed some of them.

-L152: not accounting for diffuse CO2 fluxes is an important distinction. It is an important assumption that needs to be emphasized as the natural carbon cycle will have a strong imprint in many areas.

**Response:**

We revise the sentence:

"...Therefore, we first compare the results for Paris from PMIF against those acquired based on a 3-D Eulerian atmospheric transport model by Broquet et al. (2018), the latter also accounting for uncertainties in diffuse and natural  $CO_2$  fluxes. On the one hand, the signals from these diffuse and natural  $CO_2$  fluxes cannot be modelled effectively by a Gaussian plume model. On the other hand, the diffuse and natural  $CO_2$  fluxes in Paris was shown to have only a weak impact on the inversion of fossil fuel  $CO_2$  emissions (Staufer et al., 2016). For this comparison, ..."

In addition, we add more discussions on the impact of biogenic fluxes in more detail:

Ln 519-523: "...Broquet et al. (2018) showed that systematic error could hamper the ability of the inversion system to reduce the errors in the emissions estimates. Thirdly, we neglect the impact of uncertainties in diffuse fossil fuel CO2 emissions (outside clumps) and non-fossil CO2 fluxes (within and outside clumps), the latter including net ecosystem exchange (NEE) from the terrestrial biosphere, the CO2 emitted by the burning of biofuel, the respiration from human and animals (Ciais et al., 2020) and the net CO2 fluxes between the atmosphere and ocean. For example, the signals from terrestrial NEE can be strong during the growing season, and the signals from ocean CO2 fluxes may have a critical impact on the overall XCO2 patterns in the proximity of coastlines. In principle, the signals of diffuse fossil fuel CO2 emissions and non-fossil CO2 fluxes outside the clumps can be potentially filtered by removing the local background XCO2 field to extract plumes generated only by emissions from clumps (Kuhlmann et al., 2019; Ye et al., 2020; Zheng et al., 2020a). The non-fossil

 $CO_2$  fluxes within clumps vary from clump to clump, and could contribute a non-negligible fraction of the total  $CO_2$  fluxes in many clumps (Br éon et al., 2015; Ciais et al., 2020; Wu et al., 2018). The satellite observations alone cannot effectively differentiate the fossil fuel  $CO_2$  emissions and the non-fossil  $CO_2$  fluxes within clumps. In the clumps with non-negligible non-fossil  $CO_2$  fluxes, the inversion of fossil fuel  $CO_2$  emissions could be influenced (Ye et al., 2020; Yin et al., 2019). Fourthly, ..."

-L225: a simple description of the sigma parameter (e.g., what determines it) will help the reader without having to refer to Ars et al. (2017).

**Response:**

To clarify our set-up of the parameters in the Gaussian plume model used here, we revise the sentence in Ln 225: "The  $\sigma_j$  is a function of downwind distance *i* and atmospheric stability parameter:  $\sigma_j = \beta j/(1+\gamma j)^{-1/2}$ , where  $\alpha$  is a coefficient that converts the computed XCO2 enhancement in the unit of ppm, and  $\beta$  and  $\gamma$  are coefficients depending on the atmospheric Pasquill stability category which is a function of the wind speed and solar radiation (Turner, 1970). The values for  $\beta$  and  $\gamma$  can be found in Bowers et al. (1980). The original Gaussian plume model generates a stationary plume... "

**-L369: why not just use Fig. S3 for side by side comparison?**

**Response:**

Fig. S3b is adapted from Fig. 6 in Broquet et al. (2018), Copernicus Publications. We assume it is not allowed to put it in the main text. If the editor can confirm it can be put it in the main text without any copyright issue, we agree to replace Fig. 1 with Fig. S3.

-L404: "N20". There are quite some acronyms already that need checking back and forth. Will improve the reading removing some that do not have intuitive meanings.

**Response:**

We have acronyms of "N20", "D20" for the assessment of the posterior uncertainties. We also have acronyms of "AMS", "ASS", "MCS", "SCS", "SectCS", "NoCor" for the configuration of prior uncertainty. Each acronym has a long explanation, and we found it is not easy to adapt the manuscript without using these acronyms. However, we summarize all the acronyms in an Appendix to help the readers.

**-L501: How about the optimized state? Curious how well will the Gaussian Plum model do if it assimilates the psuedo observations generated using the full 3-D models in this case. It will be a strong demonstration if it can get the emission order general variations right!**

**Response:**

As it stands, PMIF can be used to process individual samples of pseudo prior fluxes and pseudo observations and compute pseudo posterior fluxes to assess error reductions to a pseudo truth. All the numerical objects needed to apply Eq.2 are built in this system as reflected by its description. However, if the errors injected in such OSSEs with explicit pseudo data are consistent with the statistics of uncertainties know by the inversion system, the statistics of errors in the flux estimates are fully characterized by  $\mathbf{A}$  (since the observation operator is linear), whose direct computation is thus the best index of the potential of the inversion and of a given

observation network (Wang et al., 2018). This is why we only focus on such a computation here.

PMIF is mainly designed for OSSEs and would require some adaptations and extensions to process real satellite images or the pseudo observations generated by a 3-D model. For example, it requires to remove the XCO2 background concentrations underlying the detected plumes in the observations that could be assimilated by the system. More importantly, the Gaussian model may have difficulties to fit the plumes generated by a 3-D model in some cases: because of the turbulence close to the source, of the 3D variations in the wind field, and of multiple other parameters (like variations in the topography, the complexity of vertical mixing etc.). As done by Nassar et al. (2017), the wind direction might need some adjustment in some cases.

However, the difficulty of fitting the model simulation to the actual plumes sampled by the observation is also a traditional weakness in atmospheric inversion when the complex mesoscale atmospheric transport models are used; this explains why many of the recent inversions of  $CO_2/CH_4$  plant and city emissions that have been conducted based on OCO-2/TROPOMI data use Gaussian models or a Gaussian approximation of the shape of the plume to apply direct flux computations in the data (e.g. Nassar et al., 2017; Reuter et al., 2019; Zheng et al., 2020).

In addition, the study by Prunet et al. (2020) (the talk available at https://cdn.eventsforce.net/files/ef-xnn67yq56ylu/website/9/5\_734\_pascal\_prunet-

\_plume\_detection\_and\_characterization\_from\_xco\_\_imagery-

evaluation\_of\_gaussian\_methods\_for\_quantifying\_plant\_and\_city\_fluxes.pptx) indicates that Gaussian models fit the plumes from true mesoscale models well enough (so that the inversions using the Gaussian model can provide a good estimate of the emissions) for a good part of the typical atmospheric conditions encountered around the set of European cities and plants they investigated.

So we think the use of a Gaussian plume model does not bias the results discussed in the paper given the considerations listed above.

-L519: Quite a few studies explore the interfering effect of natural CO2 fluxes.

Wu, K., Lauvaux, T., Davis, K. J., Deng, A., Lopez Coto, I., Gurney, K. R. and Patarasuk, R.: Joint inverse estimation of fossil fuel and biogenic CO2 fluxes in an urban environment: An observing system simulation experiment to assess the impact of mul-tiple uncertainties, Elem Sci Anth, 6(1), 17, doi:10.1525/elementa.138, 2018.

Yin, Y., Bowman, K., Bloom, A., Worden, J.: Detection of fossil fuel emission trends in the presence of natural carbon cycle variability, Environmental Research Letter, 14(8):084050, doi:10.1088/1748-9326/ab2dd7, 2019.

**Response:**

See the response before about non-fossil CO2 fluxes.

-L538: Again, I understand that 20% posterior uncertainty is a desirable goal, but it did not provide a full picture if the values for the high emission densities are only at the order of 10 days for a year. Other references will help define the landscape.

**Response:**

As discussed above, this 20% threshold is used to quantify only the cases when the emissions are "well constrained".

In this paragraph, what we want to discuss is the large variation of N20 within each emission bin. If we choose other threshold, it does not change the fact that the clumps within each bin are not be equally constrained: the frequency of clear-sky days still largely impacted the performance of the inversion.

-Figure 3: the number of clamps is repeated in every plot from Fig. 3-5. Reductant to repeat so many times. Maybe indicate clearly that (a) and (b) are the same just for different experiments.

**Response:**

We remove the number of clumps in Figs. 3-5.

---

## Author Response (AR1)

**Response to comments on "PMIF v1.0: an inversion system to estimate the potential of satellite observations to monitor fossil fuel CO2 emissions" by Y. Wang et al.**

We thank the referee for reviewing our manuscript and for his valuable comments and suggestions. Please find attached a point-by point reply (in black) to each of the comments raised by the referee (in blue) with legible text and figures organized along the text. For your convenience, changes in the revised manuscript are highlighted with dark red. All the pages and line numbers correspond to the original version of text.

**1 Overview:**

Review of "PMIF v1.0: an inversion system to estimate the potential of satellite observations to monitor fossil fuel CO2 emissions" by Wang et al. Wang et al. present an OSSE framework to estimate error reductions for a proposed satellite. It's based on a Gaussian plume that they run for many emission hotspots. They've done this over a large domain (globally) at fairly high spatial resolution (2 km). The work is interesting but the description of the methods could use quite a bit of work. There are some important steps in the actual implementation that are quite convoluted. Fixing this seems like a critical for publication in a journal focused on geoscientific model development. I suggest major revisions for the manuscript.

**Response:**

We carefully revised our manuscript following the comments and suggestions. We think that the revised manuscript explained the steps of the method much clearer.

**2 Comments:**

**2.1 Solution to their inversion**

I'd prefer the authors not use A as the posterior covariance matrix, I usually think of A as the averaging kernel. This is particularly confusing because you are solving for emission reductions that are the diagonals of the averaging kernel matrix.

**Response:**

We are aware of that A (or AK) is used for averaging kernel in the community of satellite retrievals (Boesch et al., 2011; Cogan et al., 2012; O'Dell et al., 2012; Wu et al., 2018b; Yoshida et al., 2011). We also know that in some books on atmospheric inversion, A is used to represent "the sensitivity of the optimal estimate to the true state" and is also called averaging kernel (such as in Daniel Jacob's "Lecture on Inverse modelling" http://acmg.seas.harvard.edu/education/jacob lectures inverse modeling.pdf), where  $S_A$  and  $\hat{S}$ are used for prior and posterior uncertainty. The  $P_f$  (with f for forecast) and  $P_a$  (with a for analysis) notations from the weather data assimilation community are also sometimes used in the GHG flux inverse modeling community. But flux inversion does not involve forecast steps, so A is widely used to represent the posterior covariance matrix in massive studies on atmospheric inversion (Broquet et al., 2018; Chevallier et al., 2005; Rayner et al., 2019) and in Peylin et al. (2013) which synthetizes the contributions from a wide range of inverse modeling groups. In particular, the review on atmospheric inversions by Rayner et al. (2019) tries to build consensus in the inverse modeling community regarding the notation and encourages the use of A for posterior uncertainty covariance matrices. So in this study, we choose to follow this suggestion.

In addition, we want to clarify here we are not only solving for the diagonals of the posterior uncertainty matrix **A**. When we investigate the posterior uncertainty at daily and annual scales (Sect. 3.3 and 3.4), we account for the temporal auto-correlations in the prior uncertainty matrix **B**, which are the off-diagonals. The resulting **A** matrix is not a diagonal matrix, and we aggregate the **A** matrix at the scales of 3 h and 21 h time windows to daily and annual scales accounting for the off-diagonal entries of **A**.

In any case, Supplemental Section 1 presents what the authors are actually doing, which differs from the equations they present in Eq. 1 and 2. In Supplemental Section 1 the authors present a derivation that is both important and convoluted. It's unclear if this is something the authors devised themselves or if it follows from other work. Typically when people decompose error covariance matrices into spatial and temporal components they use a Kronecker product (e.g., Yadav & Michalak, GMD 2013). The Kronecker product greatly reduces the computational expense. The assumptions that go into a Kronecker product are also easy to follow because it is widely used. It's also amenable to sparse matrices (I'm assuming the authors are using sparse matrices). I think the authors should remove Equation 2 and bring Supplemental Section 1 into the main text. Supplemental Section 1 is important because this is what they are actually doing. This seems like the main contribution to me.

**Response:**

Eq. 1 and 2 explains the primary principle of atmospheric inversion and PMIF. We explained in Ln 143 "PMIF is an analytical inversion system that solves for Eq. (1) by building the different matrices involved in this equation." In the revised manuscript, we update this sentence with "PMIF is an analytical inversion system that solves for Eq. (1) or for an approximation of this equation (when accounting for temporal correlations in **B**) by building the different matrices involved in this equation."

PMIF attempts at solving for Eq. 1 as well as possible. The PMIF-Paris OSSE and the experiment Exp-NoCor in PMIF-Globe solve for Eq. 1. Accounting for the temporal correlation in prior uncertainties (**B**) in other experiments in PMIF-Globe prevents from applying Eq. 1, and the Supplemental Section 1 (in the revised manuscript, it will be moved to the main text) explained how an approximation of the full  $\mathbf{A}$  is derived in practice. We regularly use the Kronecker product for modeling spatio-temporal correlations, or temporal correlations at different temporal scales in inversions, e.g. in Wang et al. (2018), or to reduce the size of **B** matrices to be inverted in variational inversions, e.g. Broquet et al. (2011). But the Kronecker product cannot help to solve for the inversions of the  $\mathbf{B}^{-1} + \mathbf{M}^{T} \mathbf{R}^{-1} \mathbf{M}$  matrix whose dimension is huge (on the order of  $10^7 \times 10^7$  since the control vector consist of  $365 \times 2 \times 11,314 = 8.3 \times 10^6$ ) and whose non-diagonal terms can expand far from the diagonal when accounting for temporal correlations in PMIF-Globe inversions. In addition, due to the large number of satellite observations, **MBM**T+**R** is even larger, being  $2.7 \times 10^7$  by  $2.7 \times 10^7$ . In Yadav and Michalak (2013), they computed and inverted the full  $MBM^{T}+R$  matrix despite using the Kronecker product to gain computational efficiency for other diagnostics. But computing and inverting B-  $^{1}+\mathbf{M}^{T}\mathbf{R}^{-1}\mathbf{M}$  or  $\mathbf{MBM}^{T}+\mathbf{R}$  in PMIF would require approximately 6000 TB of RAM, which is too much for the super computers in our lab.

In addition, as explained above, we need to aggregate the posterior uncertainty matrix A

at daily and annual scales. At these scales, **A** integrates the constraints from the temporal correlations in **B** and the spatial overlapping of plumes  $\mathbf{M}^{T}\mathbf{R}^{-1}\mathbf{M}$ , and the spatial overlapping of plumes differs from day to day depending on the wind fields. We do not see that  $\mathbf{B}^{-1}+\mathbf{M}^{T}\mathbf{R}^{-1}\mathbf{M}$  or **A** is necessarily a sparse matrix that can be computed with the Kronecker product.

Therefore, we actually devised the algorithm in Supplemental Section 1 by ourselves to approximate the diagonal of the full **A**. We admit that this method does not solve for **A** exactly, but only approximates the **A** at the scales we are interested in.

To prove that this approximation is good, we conduct an experiment with the ASS configuration of prior uncertainty where the inversion period and domain are limited to 6 months and to the Benelux, a region with high emission density and in which the 95 emission clumps are close to each other (Fig. R1a). It is reasonable to assume that if the approximation of the posterior uncertainty of emissions from clumps within this region (because we ignore the filtering of information from different spatial overlaps of plumes on different days, see the method) is good, clumps outside this inversion domain will have very marginal impact on the results for the clumps in Benelux. In this case, the full **A** (Inv-fullA) to that obtained with the approach we proposed (Inv-2step). Figure R1b shows the posterior uncertainties in the emission budgets over individual time windows 8:30-11:30 for an exemplary clump (Antwerp) from the two computations. The results from the two computations are very close, except for very few days, and the aggregated uncertainty in emission budget for the whole period differ by less than 0.1%. This confirms that our method provides a good approximation of **A** at daily to annual scales for individual clumps with reasonable accuracy.

**Figure R1** a) Distribution of emission clumps in the Benelux region that we account for in the InvfullA and Inv-2step inversions. The solid lines depict the boundaries of clumps. b) Posterior uncertainty of each single 8:30-11:30 window for Antwerp clump during the first half of the year. The green dots are the results from Inv-fullA, and the circles are the results from Inv-2step.

To address the reviewer's concern, we revised the manuscript by moving Supplemental Section 1 to the main text and slightly improving it:

"In this second set of OSSEs, PMIF-Globe, we conduct inversions for all the clumps over one year. However, the large sizes of the control vector, of the observation vector and of the associated covariance matrices prevent the derivation of a full **A** for all the clumps and all the time windows using Eq. (1). In PMIF, we thus propose and apply a two-step computation that approximates Eq. (1). This computation assumes that the system has a limited capability to improve the separation between plumes from distinct clumps on a given day by crossing the information obtained from different days. In that sense, the inversion considers the uncertainty reduction obtained for individual days when considering all the clumps together (first step, see below) before focusing on individual clumps to account for temporal correlations in the prior uncertainty (the second step, see below). In other words, we assume that when crossing information between different time windows for a given clump, the impact of filtering information from different spatial overlaps of plumes on different days is relatively smaller than that of temporal auto-correlation in the prior uncertainty. It is proven that this method provides a good approximation of  $\mathbf{A}$  at daily to annual scales for individual clumps (Supplementary text S1).

In the first step, Eq. (1) is applied to each  $10 \times 10^{\circ}$  spatial inversion windows on each day separately (corresponding to an 8:30-11:30 time window for clumps within the spatial inversion windows), by using the corresponding blocks in **B**:

$$\mathbf{A}_{\text{spt,i,j}} = \left(\mathbf{B}_{\text{spt,i,j}}^{-1} + \mathbf{M}_{\text{spt,i,j}}^{T} \mathbf{R}_{\text{spt,i,j}}^{-1} \mathbf{M}_{\text{spt,i,j}}\right)^{-1}$$
(6)

Where *i* is the *i*th spatial inversion window and *j* is the *j*th day during one year. Here,  $\mathbf{B}_{\text{spt,i,j}}$  is a diagonal matrix that only contains the variances of prior uncertainties in emissions during 8:30-11:30 for the clumps within the inversion window.  $\mathbf{M}_{\text{spt,i,j}}$  accounts for the spatial overlap of plumes generated from nearby clumps. Then we derive a "instant"  $\mathbf{M}^{T}\mathbf{R}^{-1}\mathbf{M}$  (denoted as

 $\mathbf{M}_{1,\mathbf{j},\mathbf{k}}^{\mathrm{T}} \widehat{\mathbf{R}_{1,\mathbf{j},\mathbf{k}}^{-1}} \mathbf{M}_{1,\mathbf{j},\mathbf{k}}$ ) for a given clump k at each 8:30-11:30 time window:

$$\mathbf{M}_{1,j,k}^{\mathrm{T}} \widehat{\mathbf{R}_{1,j,k}^{-1}} \mathbf{M}_{1,j,k} = \left( a_{\mathrm{spt},i,j}(k)^{-1} - b_{\mathrm{spt},i,j}(k)^{-1} \right)^{-1}$$
(7)

Where  $a_{\text{spt,i,j}}(\mathbf{k})$  is a scalar from  $\mathbf{A}_{\text{spt,i,j}}$  representing the variance of posterior uncertainty of emission from clump k in *i*th spatial inversion window and in 8:30-11:30 time window on day *j* obtained by Eq. (6), and  $b_{\text{spt,i,j}}(\mathbf{k})$  is the scalar from  $\mathbf{B}_{\text{spt,i,j}}$  representing the variance of prior uncertainty for the same control variable.

In the second step, the inversion is conducted for each clump k separately, considering the correlation in time in **B**, using  $\mathbf{M}_{1,\mathbf{j},\mathbf{k}}^{\mathrm{T}} \widehat{\mathbf{R}_{1,\mathbf{j},\mathbf{k}}^{-1}} \mathbf{M}_{1,\mathbf{j},\mathbf{k}}$  derived from the first step:

$$\mathbf{A}_{tmp,k} = \left( \mathbf{B}_{tmp,k}^{-1} + \begin{bmatrix} \mathbf{M}_{i,1,k}^{\mathrm{T}} \widehat{\mathbf{R}_{i,1,k}^{-1}} \mathbf{M}_{i,1,k} & 0 & 0 \\ 0 & \ddots & 0 \\ 0 & 0 & \mathbf{M}_{i,n,k}^{\mathrm{T}} \widehat{\mathbf{R}_{i,n,k}^{-1}} \mathbf{M}_{i,n,k} \end{bmatrix} \right)^{-1}$$
(8)

Where n=366×2, representing the time windows for 8:30-11:30 and for the rest 21 hours on the 366 days of one year (2008).  $\mathbf{B}_{tmp,k}$  is the covariance matrix accounting for the temporal auto-correlation in the prior uncertainty for a single clump:

$$\mathbf{B}_{\mathrm{tmp},k} = \begin{bmatrix} \sigma_{t1}^{2} & cov(\varepsilon_{t1}, \varepsilon_{t2}) & \dots & cov(\varepsilon_{t1}, \varepsilon_{tn}) \\ cov(\varepsilon_{t1}, \varepsilon_{t2}) & \sigma_{t2}^{2} & \dots & cov(\varepsilon_{t2}, \varepsilon_{tn}) \\ \vdots & \vdots & \ddots & \vdots \\ cov(\varepsilon_{t1}, \varepsilon_{tn}) & cov(\varepsilon_{t2}, \varepsilon_{tn}) & \dots & \sigma_{tn}^{2} \end{bmatrix}$$
(9)

In PMIF-Globe, we first conduct the inversion in which the prior uncertainty has no

**temporal auto-correlation (Exp-NoCor) ... "**

Finally, I would strongly suggest not using "pseudo" in Supplemental Section 1 because that implies computing a pseudoinverse, which has a very specific mathematical definition. Unless, of course, the authors are computing a pseudoinverse in which case that should be made clear.

**Response:**

We agree that "pseudo" can be misleading. In the revised manuscript, we bring the Supplemental Section 1 in Sect. 2.7.2 and replace "pseudo  $\mathbf{M}^{T}\mathbf{R}^{-1}\mathbf{M}$ " with "instant  $\mathbf{M}^{T}\mathbf{R}^{-1}\mathbf{M}$ ".

**The authors should change the title. It's not an inversion framework as they are not estimating fluxes.**

**Response:**

As it stands, this tool can be used to process individual samples of pseudo prior fluxes and pseudo observations and compute pseudo posterior fluxes to assess error reductions to a pseudo truth. All the numerical objects needed to apply Eq.2 are built in this system as reflected by its description. However, if the errors injected in such OSSEs with explicit pseudo data are consistent with the statistics of uncertainties know by the inversion system, the statistics of errors in the flux estimates are fully characterized by  $\mathbf{A}$  (since the observation operator is linear), whose direct computation is thus the best index of the potential of the inversion and of a given observation network (Wang et al., 2018). This is why we focus on such a computation here. This computation of  $\mathbf{A}$  is actually a standard atmospheric inversion computation. Technically speaking, the PMIF can also be used to assimilate real data to produce estimate of the fluxes. Based on such considerations, the PMIF is an atmospheric inversion system like others so we wish to keep this label for clarity.

Of note is that this tool is mainly designed for OSSEs and would require some adaptations and extensions to process real satellite images over the period of data availability, to remove the XCO2 background concentrations underlying the detected plumes, and maybe to more properly cope with errors in the modeling of the plumes (see our answer to the next comment) than just inflating the **R** matrix. However, such a limited account for model error in real experimental conditions is a traditional weakness of atmospheric inversion systems and other systems mainly designed for OSSEs that have always been named atmospheric inversion systems (Rayner et al., 2014; Wu et al., 2016).

**2.2 Justification on the use of a Gaussian plume**

Real plumes are only Gaussian in the time-averaged sense. The satellite observations provide a snapshot in time that likely would not be Gaussian. I think the authors need to provide some justification as to why a Gaussian plume is appropriate for data that is not time-averaged. A Gaussian plume may give a reasonable upper bound on the uncertainty reduction, but will likely induce systematic biases if implemented operationally. These potential biases should be discussed.

**Response:**

We agree with the reviewer that instant image of real plumes do not always follow a Gaussian shape: because of the turbulence close to the source, of the 3D variations in the wind

field, especially over the long distances, and of multiple other parameters (like variations in the topography, the complexity of vertical mixing etc.). However:

1) we stress, again, that the PMIF was not designed to process real data, but for OSSEs. The primary driver of the scores of posterior uncertainties and of uncertainty reduction in the PMIF which are the target of the OSSEs and of this system is the spatial extent and amplitude of the plumes, and the parameters of the Gaussian model in the PMIF are such that they fairly reproduce those from more complex models. This had been checked based on our comparisons between the results from the PMIF-Paris and from Broquet et al. (2018) as explained in the Section 3.1 and in the supplementary material.

2) the complex variations in real plumes that make them hardly Gaussian also hamper their modeling with complex mesoscale atmospheric transport models; this explains why many of the recent inversions of  $CO_2/CH_4$  plant and city emissions that have been conducted based on OCO-2/TROPOMI data use Gaussian models or a Gaussian approximation of the shape of the plume to apply direct flux computations in the data (Nassar et al., 2017; Reuter et al., 2019; Zheng et al., 2020).

3) The study by Prunet et al. (2020) (the talk available at https://cdn.eventsforce.net/files/ef-xnn67yq56ylu/website/9/5\_734\_pascal\_prunet-

plume\_detection\_and\_characterization\_from\_xco\_\_imagery-

evaluation of gaussian methods for quantifying plant and city fluxes.pptx) even indicates that Gaussian models fit the plumes from "true" mesoscale models well enough (so that the inversions using the Gaussian model can provide a good estimate of the emissions) for a good part of the typical atmospheric conditions encountered around the set of European cities and plants they investigated.

The choice of the Gaussian plume model in the PMIF was definitely linked to its light computation cost while using 2 km resolution observations and solving for emissions at a high resolution across the globe and a year. We think this choice does not bias the results given the different considerations listed above.

To better address this discussion about the Gaussian plume model in the manuscript, we revised it by:

1) revising Ln 101: "Therefore, in this study, we develop a Plume Monitoring Inversion Framework (PMIF) and conduct a set of Observing System Simulation Experiments (OSSEs) to assess, for the first time, the performance of a satellite instrument to monitor the emissions of all the clumps across the globe and over a whole year. The imager studied has the foreseen characteristics of the individual satellites of the forthcoming CO2M mission. It would be a high-resolution spectrometer, with 2 km  $\times$  2 km resolution pixels and a swath of 300 km, and it would be placed on a sun-synchronous orbit ensuring global coverage in 4 days. The PMIF inversion system relies on the list of clumps extracted by Wang et al. (2019) from the ODIAC inventory, on a Gaussian plume model to simulate the XCO2 plumes generated by the emissions from these clumps, on an analytical inverse modeling framework, and on a combination of overlapping assimilation windows to solve for the inversion problem over the globe and a full year. It also addresses the question of temporal extrapolation that is needed to generate estimates of annual emissions from the information of a limited number of time windows for which emissions are well constrained by the direct satellite images, by accounting for the temporal auto-correlation of the prior uncertainties. The performance is assessed in terms of the

uncertainties in the emissions (Sect. 2.1) at different scales. The PMIF uses a Gaussian plume model at the local scale to ensure that the computation cost is affordable. Such a model can often hardly fit with actual plumes over the distances considered in this study (due to variations in the wind field, topography, vertical mixing etc. over such distances) but is shown, when driven with suitable parameters, to provide a satisfactory simulation of the plume extent and amplitudes, which appear to be the main drivers of the targeted computations of uncertainties in the emissions in our OSSE framework (as shown in section 3.1). In PMIF, we also ignore the impact of some sources of uncertainties on the inversion of emissions, including systematic errors on the XCO2 retrievals, the impact of uncertainties in diffuse anthropogenic emissions outside clumps, in non-fossil CO2 fluxes (within and outside clumps), and in the spatial and temporal variations of emissions within the clump and the short time windows that the inversion aims to solve. These impacts are discussed in detail afterwards."

2) revising Ln 148-157: "We use a Gaussian plume model (Sect. 2.4) to simulate the atmospheric transport at a spatial resolution consistent with that of the XCO2 measurements from the planned CO2 imager and with the highly heterogeneous distribution of emissions. Compared with complex 3-D atmospheric transport models, Gaussian plume models have a very low computational cost, making the global assessment of posterior uncertainty and uncertainty reduction at the scale of emissions clumps from the assimilation of high resolution data feasible. However, since a Gaussian plume model provides a highly simplified approximation of the atmospheric transport from emission clumps, we need to verify that its use in the PMIF yields estimates of the uncertainties in the inverted emissions that are consistent with those that would be based on more complex models. Therefore, we first compare the results for Paris from PMIF against those acquired based on a 3-D Eulerian mesoscale atmospheric transport model by Broquet et al. (2018)... "

The authors should give more explanation of  $\sigma j$ . There are two parameters in a Gaussian plume model and they spend one line talking about  $\sigma j$ : "The  $\sigma j$  is a function of downwind distance i and atmospheric stability parameter. We take the form for  $\sigma j$  from Ars et al. (2017).".

**Response:**

To clarify our set-up of the parameters in the Gaussian plume model used here, we revise the sentence in Ln 225: "The  $\sigma_j$  is a function of downwind distance *i* and atmospheric stability parameter:  $\sigma_j = \beta j/(1+\gamma j)^{-1/2}$ , where  $\alpha$  is a coefficient that converts the computed XCO2 enhancement in the unit of ppm, and  $\beta$  and  $\gamma$  are coefficients depending on the atmospheric Pasquill stability category which is a function of the wind speed and solar radiation (Turner, 1970). The values for  $\beta$  and  $\gamma$  can be found in Bowers et al. (1980). The original Gaussian plume model generates a stationary plume... "

**2.3 Clumps**

I don't like the terminology "emission clumps". It doesn't fit with the actual definition of a clump:

noun: "a compacted mass or lump of something"

verb: "form into a clump or mass"

Emissions don't clump. The various sources have just been grouped together. The abstract of

their 2019 paper seemed to use "hotspot" and "clusters" which I would prefer to "clump". A cluster would be a much more intuitive name for this.

**Response:**

In our 2019 paper (Wang et al., 2019), we used the word "emission clump", which was defined as "clusters of emitting pixels (called emission clumps hereafter) that will generate individual  $XCO_2$  plumes that are detectable from space". Since we strongly link our paper to Wang et al. (2019), we believe, for clarity and consistency, that keeping the term "clump" is critical.

We can also mention that in Merriam-Webster's Collegiate Dictionary, one of the definition given for "clump" is "a group of things clustered together" (https://www.merriam-webster.com/dictionary/clump). So we think "clump" is still appropriate, in the context of American English.

**2.4 References**

The authors show a very strong bias towards European studies. They don't seem to mention any of Ray Nasser's work in the intro even though his 2017 GRL paper used a Gaussian plume model with satellite observations to study individual sources. They also seem to have missed Eric Kort's work using GOSAT to study megacities (Kort et al., GRL 2012; among others).

**Response:**

We thank the reviewer to remind these references. In the revised introduction, we rewrite the paragraph setting the context for XCO2 plume inversions:

Ln 55 "... Alternatively, vertically integrated columns of dry-air mole fractions of CO2 (XCO2) from satellites offer the opportunity to sample the atmosphere with a global coverage. Kort et al. (2012) and Janardanan (2016) found that significant  $XCO_2$  enhancements could be detected over some megacities using Greenhouse Gases Observing Satellite (GOSAT) XCO2 observations. Schwandner et al. (2017) also found  $XCO_2$  enhancements of 4.4 to 6.1 ppm in the Los Angeles urban CO2 dome using observations from Orbiting Carbon Observatory-2 (OCO-2). Nassar et al. (2017) used the  $XCO_2$  observations from OCO-2 to quantify  $CO_2$ emissions from several middle- to large-sized coal power plants. However, the design of GOSAT and OCO-2 observations with sparse sampling was mainly focused on the monitoring of CO2 natural fluxes. Recent studies show a limited amount of clear detections of transects of XCO2 plumes from cities or plants in OCO-2 observations (Zheng et al., 2020) so that GOSAT and OCO-2 data keep on being hardly used to estimate  $CO_2$  city emissions. The potential for reducing uncertainties in fossil fuel CO2 emissions at the scale of point sources (Bovensmann et al., 2010), cities (Broquet et al., 2018; Pillai et al., 2016) and agglomerations of several cities (O'Brien et al., 2016) should dramatically change with the planned satellite missions with imaging capabilities. These studies consistently showed that ..."

**2.5 3 hours vs 6 hours**

Why is there a 6-hour window for Paris and a 3-hour window globally? I see, it's defined afterward. This should be moved forward to explain why Broquet chose 6 hours and why they choose 3 hours. How is 3 hours chosen? It seems to just be picked randomly.

**Response:**

Broquet et al. (2018) showed that the XCO2 signature of the emissions from Paris is hardly detectable after 6 hours due to atmospheric diffusion, and they thus only inverted emissions during the 6 h before satellite overpasses. In PMIF-Paris experiments, we aim to compare the performance of inversion system using a Gaussian plume model with the one using a 3-D Eulerian atmospheric transport model, so we choose the same time length as Broquet et al. (2018) for PMIF-Paris. For PMIF-Globe, we already explained in the manuscript (in the revised version, we bring the explanation to Sect. 2.1, see below). On the other hand, three hours is the typical time scale that Nassar et al. (2017) used to interpret the results from their inversion of emissions from coal power plants using OCO-2 observations with a Gaussian plume model.

In the revised manuscript, we bring the explanation about the 6-hour time window for PMIF-Paris and 3-hour time window for PMIF-Globe to Sect. 2.1:

Ln 157: "Table 1 and 2 summarize the different options for the configuration of the system and of the OSSEs. One distinction between PMIF-Paris and PMIF-Globe is that PMIF-Paris relates XCO2 signals with the mean emissions 6 hours before overpasses, while it is assumed that in PMIF-Globe that the XCO2 signals only provide effective constraints on 3 h mean emissions before individual overpasses. The 6-hour period corresponds to the period of emissions from Paris whose signature in the XCO2 field can still be detected by the satellite despite the atmospheric diffusion (Broquet et al., 2018). While Broquet et al. (2018) indicated that the period of "detectable" emissions from a large megacity like Paris could last up to 6hours, most of the clumps across the globe have smaller emission rates than Paris, or are located in more complex environment close to other major emission areas where XCO2 signals can be attributed to multiple sources, making the detection of the XCO2 signature of emissions few hours before the satellite overpass even more difficult. For the PMIF-Globe experiments, we thus conservatively assume that the XCO2 signals can only provide effective constraints on 3 h mean emissions before individual overpasses in general."

We also rewrote the paragraph in Sect. 2.3:

Ln 179-Ln186: "In the PMIF-Paris inversion, the satellite observations are sampled at 11:00 local time, in line with the experiments from Broquet et al. (2018). The inversion solves for the mean emissions for the 6 hours before 11:00 local time. Broquet et al. (2018) solved for the hourly emissions during this 6-hour period but PMIF can only solve for the mean emissions during the 6-hour period due to the fact that the Gaussian plume model cannot be used to compute the signatures in the XCO2 field of individual hourly emissions during that period. The control parameter in PMIF-Paris for each overpass (Sect. 2.7.1) is thus a scaling factor  $\lambda$  for the mean emission between 05:00 and 11:00 ..."

**3 Specific comments:**

Title: Remove fossil fuel from the title. I don't see how they could differentiate fossil from non-fossil sources in their analysis.

**Response:**

In this study, all the inversions and discussions focus on fossil fuel  $CO_2$  emissions since this should be the main target of  $CO_2$  emission monitoring systems, and since the PMIF is based on an inventory of these emissions and assumes that uncertainties in other fluxes weakly impact the inversion of these emissions in clumps. However, we agree that the separation between fossil fuel emissions and non-fossil  $CO_2$  fluxes is a critical topic for the space-borne (and more generally atmospheric) monitoring of the fossil fuel emissions. Firstly, background concentrations around the plumes from fossil fuel emission clumps might be sometimes difficult to properly separate (Kuhlmann et al., 2019). This background consists in a mix of the signature of all kind of  $CO_2$  fluxes outside or within the clump boundaries. However, in a general way, uncertainties in this background can be seen as a source of uncertainty in the estimate of the fossil fuel emissions that does not prevent us from computing the fossil fuel emissions separately. Secondly, if focusing on sources and sinks collocated with the fossil fuel emissions for cities, the separation of fossil fuel emissions from biofuel emissions, human respiration and potentially natural fluxes specific to urban areas (i.e. highly different from natural fluxes at larger scale) can definitely be difficult. We investigated some estimates of the contribution of non-fossil CO2 fluxes to the total CO2 fluxes from cities. The contribution of non-fossil  $CO_2$  fluxes to the total  $CO_2$  fluxes varies a lot from city to city and from day to day. For example, in **Î**e-de-France, the biogenic fluxes are usually considered to have small impact on the signals of fossil fuel CO2 emissions in autumn and winter, while they could become nonnegligible in summer (Br éon et al., 2015; Lian et al., 2019; Staufer et al., 2016); The biogenic CO2 fluxes could represent 5% of the total signals in Indianapolis, Indiana, U.S.A. (Turnbull et al., 2015) during winter time; Miller et al. (2018) estimated that biogenic CO2 fluxes could contribute to 25% of the total CO2 enhancement in the Los Angeles Basin based on atmospheric radiocarbon measurements; Ye et al. (2020) estimated the contribution of total XCO2 enhancement due to biogenic fluxes can be as large as  $32 \pm 27\%$  (1 $\sigma$ ) and  $24 \pm 18\%$  (1 $\sigma$ ) in winter and summer. All these estimates include the urban and rural areas, while the emission clumps defined in Wang et al. (2019) only include the areas with fossil fuel CO2 emissions being high enough to form detectible XCO2 plumes through atmospheric transport. Most of these areas are built-up areas, so the contribution of non-fossil CO2 fluxes to the total fluxes should be much smaller than the whole-city estimates as mentioned above. This can be illustrated by Fig. 4a in Lian et al. (2019) of the small biogenic fluxes in the city center of Paris and by Fig. 1 in Ye et al. (2020) of the green vegetation fraction. We thus assume that in these clump areas, the fossil fuel CO2 emissions dominate the total CO2 fluxes.

In summary, we do agree with the reviewer that the satellite observations alone do not separate the fossil fuel emissions and non-fossil fuel fluxes within or around emission clumps and that these non-fossil fuel fluxes can be non-negligible. However, as shown by previous studies, the impact of non-fossil sources is within the overall uncertainty of the estimates of emissions from real data (Reuter et al., 2019; Zheng et al., 2020).

In the revised manuscript, we discussed the impact of non-fossil fluxes in more detail:

Ln 519-523: "...Broquet et al. (2018) showed that systematic error could hamper the ability of the inversion system to reduce the errors in the emissions estimates. Thirdly, we neglect the impact of uncertainties in diffuse fossil fuel CO2 emissions (outside clumps) and non-fossil CO2 fluxes (within and outside clumps), the latter including net ecosystem exchange (NEE) from the terrestrial biosphere, the CO2 emitted by the burning of biofuel, the respiration from human and animals (Ciais et al., 2020) and the net CO2 fluxes between the atmosphere and ocean. For example, the signals from terrestrial NEE can be strong during the growing season, and the signals from ocean CO2 fluxes may have a critical impact on the overall XCO2 patterns in the proximity of coastlines. In principle, the signals of diffuse fossil fuel CO2 emissions and non-fossil CO2 fluxes outside the clumps can be potentially filtered by removing

the local background XCO2 field to extract plumes generated only by emissions from clumps (Kuhlmann et al., 2019; Reuter et al., 2019; Ye et al., 2020; Zheng et al., 2020). The non-fossil CO2 fluxes within clumps vary from clump to clump, and could contribute a non-negligible fraction of the total CO2 fluxes in many clumps (Br éon et al., 2015; Ciais et al., 2020; Wu et al., 2018a). The satellite observations alone cannot effectively differentiate the fossil fuel CO2 emissions and the non-fossil CO2 fluxes within clumps. In the clumps with non-negligible non-fossil CO2 fluxes, the inversion of fossil fuel CO2 emissions could be influenced (Ye et al., 2020; Yin et al., 2019). Fourthly, ..."

Section 2.1: Should reference the sections that define the error covariance parameters.

**Response:**

We revised the manuscript:

- Ln 145: "We characterize B, R and A by the corresponding standard deviations (σ) of uncertainty in individual or aggregations of control parameters and by the temporal auto-correlations of the uncertainties (Sect. 2.6). In the following, ...";
- Ln 154-157: "... Then we apply the system to all the emission clumps over the globe and over 1 year using a different control vector and a simulation of the XCO2 sampling by a single CO2M satellite (Sect. 2.2). The inversions for all emission clumps over the globe are called PMIF-Globe. In PMIF-Globe, we first investigate the potential of satellite observations in constraining emissions from individual days (ExpNoCor in Sect. 2.6). Then we assess the ability of satellite observations to constrain emissions at annual scale by accounting for the temporal auto-correlation of the prior uncertainties (other experiments in Sect. 2.6). Table 1 and 2 summarize the different options for the configuration of the system and of the OSSEs."

**Line 126: what is yfixed?**

**Response:**

We revised the sentence:

"... The inversion derives a statistical estimate for a set of control variables x in a model  $x \rightarrow y=Mx$  that simulates the satellite XCO2 measurements  $y^{\circ}$ . The model M linking x and y is a combination of flux and atmospheric transport models (detailed in Sect. 2.4), and is called observation operator hereafter. As explained below, we do not have a constant term added to Mx in the observation operator of the PMIF that would gather the atmospheric CO2 signature of the fluxes not controlled by the inversion (like non-fossil fluxes and the background XCO2 field) since the uncertainty in such fluxes is ignored. The inversion follows a Bayesian statistical framework,..."

Line 181: rephrase, too colloquial: "but the PMIF can hardly handle hourly emissions when covering a whole year".

**Response:**

We revised the sentence:

"...Broquet et al. (2018) solved for the hourly emissions during this 6-hour period but PMIF can only solve for the mean emissions during the 6-hour period due to the fact that the Gaussian plume model cannot be used to compute the signatures in the  $XCO_2$  field of individual

hourly emissions during that period. The control parameter for each overpass ..."

**References**

Boesch, H., Baker, D., Connor, B., Crisp, D. and Miller, C.: Global Characterization of CO2 Column Retrievals from Shortwave-Infrared Satellite Observations of the Orbiting Carbon Observatory-2 Mission, Remote Sensing, 3(2), 270–304, doi:10.3390/rs3020270, 2011.

Bowers, J. F., Bjorklund, J. R., Cheney, C. S. and Schewe, G. J.: Industrial source complex (ISC) dispersion model user's guide, US Environmental Protection Agency, Office of Air Quality Planning and Standards., 1980.

Bréon, F. M., Broquet, G., Puygrenier, V., Chevallier, F., Xueref-Remy, I., Ramonet, M., Dieudonné, E., Lopez, M., Schmidt, M., Perrussel, O. and Ciais, P.: An attempt at estimating Paris area CO2 emissions from atmospheric concentration measurements, Atmospheric Chemistry and Physics, 15(4), 1707–1724, doi:10.5194/acp-15-1707-2015, 2015.

Broquet, G., Chevallier, F., Rayner, P., Aulagnier, C., Pison, I., Ramonet, M., Schmidt, M., Vermeulen, A. T. and Ciais, P.: A European summertime CO2 biogenic flux inversion at mesoscale from continuous in situ mixing ratio measurements, J. Geophys. Res., 116(D23), doi:10.1029/2011jd016202, 2011.

Broquet, G., Bréon, F.-M., Renault, E., Buchwitz, M., Reuter, M., Bovensmann, H., Chevallier, F., Wu, L. and Ciais, P.: The potential of satellite spectro-imagery for monitoring CO2 emissions from large cities, Atmos. Meas. Tech., 11(2), 681–708, doi:10.5194/amt-11-681-2018, 2018.

Chevallier, F., Fisher, M., Peylin, P., Serrar, S., Bousquet, P., Bréon, F. M., Chédin, A. and Ciais, P.: Inferring CO2 sources and sinks from satellite observations: Method and application to TOVS data, J. Geophys. Res., 110(D24), doi:10.1029/2005jd006390, 2005.

Ciais, P., Wang, Y., Andrew, R., Bréon, F.-M., Chevallier, F., Broquet, G., Nabuurs, G.-J., Peters, G., McGrath, M., Meng, W., Zheng, B. and Tao, S.: Biofuel burning and human respiration bias on satellite estimates of fossil fuel CO2 emissions, Environ. Res. Lett., doi:10.1088/1748-9326/ab7835, 2020.

Cogan, A. J., Boesch, H., Parker, R. J., Feng, L., Palmer, P. I., Blavier, J.-F. L., Deutscher, N. M., Macatangay, R., Notholt, J., Roehl, C., Warneke, T. and Wunch, D.: Atmospheric carbon dioxide retrieved from the Greenhouse gases Observing SATellite (GOSAT): Comparison with ground-based TCCON observations and GEOS-Chem model calculations, Journal of Geophysical Research: Atmospheres, 117(D21), doi:10.1029/2012JD018087, 2012.

Janardanan, R., Maksyutov, S., Oda, T., Saito, M., Kaiser, J. W., Ganshin, A., Stohl, A., Matsunaga, T., Yoshida, Y. and Yokota, T.: Comparing GOSAT observations of localized CO2 enhancements by large emitters with inventory-based estimates, Geophys. Res. Lett., 43(7), 2016GL067843, doi:10.1002/2016GL067843, 2016.

Kort, E. A., Frankenberg, C., Miller, C. E. and Oda, T.: Space-based observations of megacity carbon dioxide, Geophys. Res. Lett., 39(17), L17806, doi:10.1029/2012GL052738, 2012.

Kuhlmann, G., Broquet, G., Marshall, J., Clément, V., Löscher, A., Meijer, Y. and Brunner, D.: Detectability of CO2 emission plumes of cities and power plants with the Copernicus Anthropogenic CO2 Monitoring (CO2M) mission, Atmospheric Measurement Techniques Discussions, 1–35, doi:https://doi.org/10.5194/amt-2019-180, 2019.

Lian, J., Bréon, F.-M., Broquet, G., Zaccheo, T. S., Dobler, J., Ramonet, M., Staufer, J., Santaren, D., Xueref-Remy, I. and Ciais, P.: Analysis of temporal and spatial variability of atmospheric CO2 concentration within Paris from the GreenLITETM laser imaging experiment, Atmospheric Chemistry and Physics, 19(22), 13809–13825, doi:https://doi.org/10.5194/acp-19-13809-2019, 2019.

Miller, J. B., Lehman, S., Verhulst, K. R., Miller, C. E., Duren, R. M., Yadav, V., Newman, S. and Sloop, C.: Unexpected and significant biospheric CO2 fluxes in the Los Angeles Basin indicated by atmospheric radiocarbon (14CO2), in 45th Global Monitoring Annual Conference, vol. 2018, pp. A53F–03, Boulder, Colorado, USA., 2018.

Nassar, R., Hill, T. G., McLinden, C. A., Wunch, D., Jones, D. B. A. and Crisp, D.: Quantifying CO2 Emissions From Individual Power Plants From Space, Geophys. Res. Lett., 44, 10045–10053, doi:10.1002/2017GL074702, 2017.

O'Dell, C. W., Connor, B., Bösch, H., O'Brien, D., Frankenberg, C., Castano, R., Christi, M., Eldering, D., Fisher, B., Gunson, M., McDuffie, J., Miller, C. E., Natraj, V., Oyafuso, F., Polonsky, I., Smyth, M., Taylor, T., Toon, G. C., Wennberg, P. O. and Wunch, D.: The ACOS CO2 retrieval algorithm – Part 1: Description and validation against synthetic observations, Atmos. Meas. Tech., 5(1), 99–121, doi:10.5194/amt-5-99-2012, 2012.

Peylin, P., Law, R. M., Gurney, K. R., Chevallier, F., Jacobson, A. R., Maki, T., Niwa, Y., Patra, P. K., Peters, W., Rayner, P. J., Rödenbeck, C., van der Laan-Luijkx, I. T. and Zhang, X.: Global atmospheric carbon budget: results from an ensemble of atmospheric CO2 inversions, Biogeosciences, 10, 6699–6720, doi:10.5194/bg-10-6699-2013, 2013.

Rayner, P. J., Utembe, S. R. and Crowell, S.: Constraining regional greenhouse gas emissions using geostationary concentration measurements: a theoretical study, Atmospheric Measurement Techniques, 7(10), 3285–3293, 2014.

Rayner, P. J., Michalak, A. M. and Chevallier, F.: Fundamentals of data assimilation applied to biogeochemistry, Atmospheric Chemistry and Physics, 19(22), 13911–13932, doi:https://doi.org/10.5194/acp-19-13911-2019, 2019.

Reuter, M., Buchwitz, M., Schneising, O., Krautwurst, S., O'Dell, C. W., Richter, A., Bovensmann, H. and Burrows, J. P.: Towards monitoring localized CO2 emissions from space: co-located regional CO2 and NO2 enhancements observed by the OCO-2 and S5P satellites, Atmospheric Chemistry and Physics, 19(14), 9371–9383, doi:https://doi.org/10.5194/acp-19-

9371-2019, 2019.

Schwandner, F. M., Gunson, M. R., Miller, C. E., Carn, S. A., Eldering, A., Krings, T., Verhulst, K. R., Schimel, D. S., Nguyen, H. M., Crisp, D., O'Dell, C. W., Osterman, G. B., Iraci, L. T. and Podolske, J. R.: Spaceborne detection of localized carbon dioxide sources, Science, 358(6360), eaam5782, doi:10.1126/science.aam5782, 2017.

Staufer, J., Broquet, G., Bréon, F.-M., Puygrenier, V., Chevallier, F., Xueref-Rémy, I., Dieudonné, E., Lopez, M., Schmidt, M., Ramonet, M., Perrussel, O., Lac, C., Wu, L. and Ciais, P.: The first 1-year-long estimate of the Paris region fossil fuel CO2 emissions based on atmospheric inversion, Atmos. Chem. Phys., 16(22), 14703–14726, doi:10.5194/acp-16-14703-2016, 2016.

Turnbull, J. C., Sweeney, C., Karion, A., Newberger, T., Lehman, S. J., Tans, P. P., Davis, K. J., Lauvaux, T., Miles, N. L., Richardson, S. J. and others: Toward quantification and source sector identification of fossil fuel CO2 emissions from an urban area: Results from the INFLUX experiment, J. Geophys. Res., 120(1), 292–312, 2015.

Turner, D. B.: Workbook Of Atmospheric Dispersion Estimeates. Office of Air Program Pub. No. AP-26, Environmental protection agency, USA., 1970.

Wang, Y., Broquet, G., Ciais, P., Chevallier, F., Vogel, F., Wu, L., Yin, Y., Wang, R. and Tao, S.: Potential of European 14CO2 observation network to estimate the fossil fuel CO2 emissions via atmospheric inversions, Atmos. Chem. Phys., 18(6), 4229–4250, doi:10.5194/acp-18-4229-2018, 2018.

Wang, Y., Ciais, P., Broquet, G., Bréon, F.-M., Oda, T., Lespinas, F., Meijer, Y., Loescher, A., Janssens-Maenhout, G., Zheng, B., Xu, H., Tao, S., Gurney, K. R., Roest, G., Santaren, D. and Su, Y.: A global map of emission clumps for future monitoring of fossil fuel CO2 emissions from space, Earth System Science Data, 11(2), 687–703, doi:https://doi.org/10.5194/essd-11-687-2019, 2019.

Wu, K., Lauvaux, T., Davis, K. J., Deng, A., Coto, I. L., Gurney, K. R. and Patarasuk, R.: Joint inverse estimation of fossil fuel and biogenic  $CO_2$  fluxes in an urban environment: An observing system simulation experiment to assess the impact of multiple uncertainties, Elem Sci Anth, 6(1), 17, doi:10.1525/elementa.138, 2018a.

Wu, L., Broquet, G., Ciais, P., Bellassen, V., Vogel, F., Chevallier, F., Xueref-Remy, I. and Wang, Y.: What would dense atmospheric observation networks bring to the quantification of city CO2 emissions?, Atmos. Chem. Phys., 16(12), 7743–7771, doi:10.5194/acp-16-7743-2016, 2016.

Wu, L., Hasekamp, O., Hu, H., Landgraf, J., Butz, A., aan de Brugh, J., Aben, I., Pollard, D. F., Griffith, D. W. T., Feist, D. G., Koshelev, D., Hase, F., Toon, G. C., Ohyama, H., Morino, I., Notholt, J., Shiomi, K., Iraci, L., Schneider, M., Mazière, M. de, Sussmann, R., Kivi, R., Warneke, T., Goo, T.-Y. and Té, Y.: Carbon dioxide retrieval from OCO-2 satellite observations using the RemoTeC algorithm and validation with TCCON measurements, Atmospheric

Measurement Techniques, 11(5), 3111-3130, doi:https://doi.org/10.5194/amt-11-3111-2018, 2018b.

Ye, X., Lauvaux, T., Kort, E. A., Oda, T., Feng, S., Lin, J. C., Yang, E. G. and Wu, D.: Constraining Fossil Fuel CO2 Emissions From Urban Area Using OCO-2 Observations of Total Column CO2, Journal of Geophysical Research: Atmospheres, 125(8), e2019JD030528, doi:10.1029/2019JD030528, 2020.

Yin, Y., Bowman, K., Bloom, A. A. and Worden, J.: Detection of fossil fuel emission trends in the presence of natural carbon cycle variability, Environ. Res. Lett., 14(8), 084050, doi:10.1088/1748-9326/ab2dd7, 2019.

Yoshida, Y., Ota, Y., Eguchi, N., Kikuchi, N., Nobuta, K., Tran, H., Morino, I. and Yokota, T.: Retrieval algorithm for CO2 and CH4 column abundances from short-wavelength infrared spectral observations by the Greenhouse gases observing satellite, Atmos. Meas. Tech., 4(4), 717–734, doi:10.5194/amt-4-717-2011, 2011.

Zheng, B., Chevallier, F., Ciais, P., Broquet, G., Wang, Y., Lian, J. and Zhao, Y.: Observing carbon dioxide emissions over China's cities and industrial areas with the Orbiting Carbon Observatory-2, Atmospheric Chemistry and Physics, 20(14), 8501–8510, doi:https://doi.org/10.5194/acp-20-8501-2020, 2020.

**Response to comments on "PMIF v1.0: an inversion system to estimate the potential of satellite observations to monitor fossil fuel CO2 emissions" by Y. Wang et al.**

We thank the referee for reviewing our manuscript. Please find attached a point-by point reply (in black) to each of the comments raised by the referee (in blue) with legible text and figures organized along the text. For your convenience, changes in the revised manuscript are highlighted with dark red. All the pages and line numbers correspond to the original version of text.

This study assesses the potential of satellite imagery of a future mission CO2M XCO2 to constrain the emissions from cities and power plants over the whole globe for one year. To reduce the computational cost of the traditionally used 3-D full transport models, this study simplified the observation operator with a few idealized hypotheses: (a) a Gaussian plume model, no model errors, (b) no overlapping effects from nearby hotspots, (c) no impact of natural carbon cycle fluxes. It is useful to get a global-scale estimate for the potential of emission uncertainty reductions for the proposed mission – even though the results are not very positive in terms of  $CO_2$  measurements' potential in constraining fossil fuel  $CO_2$  emissions alone given those idealized setups.

**Response:**

We would like to clarify the point (b) listed above by the reviewer. Actually, there can be some overlapping between the plumes generated by nearby clumps in the PMIF. In Eulerian transport model, the plumes from nearby sources can converge along atmospheric circulation. However, here, since using a classical Gaussian plume model, the plumes are straight along the wind direction. Therefore, the plumes from two nearby clumps can cross each other, but they'll systematically diverge on long distances, which, in some cases, can lead to a significant underestimation of the plume overlapping. To make it clearer, we revised the sentences Ln 506-508: "...Firstly, the plumes generated by the Gaussian plume model are straight along the wind direction at the source pixel. As a result, we allow the plumes from nearby clumps to potentially cross each other, but these plumes overlapping along the atmospheric circulation like Eulerian transport models. In this sense, the overlapping effect of plumes can be underestimated in PMIF. In a realistic situation of atmospheric transport, if plumes from multiple clumps overlap very often, the inversion performance for individual clumps will be degraded since it will have the difficulties to accurately attribute the XCO2 signals to individual clumps."

**General comments:**

The authors highlight the global scope of this study, but no global distribution is shown. Fig. 6 shows information about US and China, why only these two regions? The global results are aggregated with emission density bins (Fig 2 - 5), which I assume is not the only determining factor. With simple statistics of median spread, a lot of information is lost. It does not really provide a "global" view. Fig. 1 highlighted the impacts of wind speed, which may create spatial patterns that overlay with emission density maps. Such information may reveal a better global overview.

**Response:**

We synthesize the global results with the plot of median values and the spread in Figs. 2-5. Figure 6 is shown to prove that the frequency of clear-sky largely explains the large variations within each emission bin. We agree with the reviewer that the inversion results are mainly driven by a combination of emission rates, wind speed and frequency of clear-sky days. However, plotting clumps' uncertainty on top of clump emissions or wind speed would make the figure too saturated to read. (Figure 6c and d are already close to a saturation of dots). Following the reviewer advice, we have produced figures like Figure 6c and d for all the regions of the globe. However, since they do not bring much more qualitative insights than Figure 6c and d, we have put them in the supplementary material. In the main text, we remind the readers to refer to these figures accordingly:

Ln 410: "At regional scale (Figs. S4, S5), South America, North America, and Africa tend to have larger N20 values for same bin of clump annual emission than the other regions, while Middle East and Asia have the lowest ones. In addition, there are large variations and spatial heterogeneity in the N20 values within each emission bins (Fig. S5), which will be further discussed in Sect. 4."

Ln 545: "... These results illustrate the dependence of the potential of satellite observations to constrain emissions on the frequency of clear-sky conditions. The relative uncertainty in the inversion of the emissions from a clump is primarily driven by the budget of these emissions, and by the wind speed (as illustrated by Fig. 1). The frequency of clear-sky days modulates the number of direct observation of the plume from a clump and thus the number of days for which the inversion can decrease the uncertainty when ignoring temporal auto-correlations in the prior uncertainty in Exp-NoCor. The frequency of clear-sky day, together with the emission rate and wind speed, are the main drivers of the posterior uncertainty in daily to annual emissions when accounting for temporal auto-correlations in the prior uncertainty."

Also, a posterior uncertainty of 20% has been used as a benchmark throughout the paper (given a 30% prior uncertainty). However, only a few cases/days can meet such a requirement. Thus, it may be more helpful to show what posterior uncertainty can be achieved for a given length of days across typical regions (e.g., using a 2-D matrix?)

**Response:**

Firstly, we stress that the prior uncertainties are different at different time scale. In all the experiments, the prior uncertainty is 30% for annual emissions. When decomposing the uncertainty of annual emissions to the scales of 3 h and 21 h time windows, the resulting uncertainties largely depend on the assumption about the temporal auto-correlations (Sect. 2.6). In the ASS scenario, the prior uncertainty for 3 h emissions is  $\sqrt{(44\%^2+26\%^2)}=51\%$ , while in NoCor scenario, it is 614%.

Eq. (1) shows that the posterior uncertainty and uncertainty reduction depend on the prior uncertainty. For example, if the projection of uncertainties in satellite observations on the uncertainty in emissions (i.e.  $\mathbf{M}^{T}\mathbf{R}^{-1}\mathbf{M}$ ) equals to 50% for a single 3 h time window, in ASS scenario, the posterior uncertainty equals to  $\sqrt{1/(1/(51\%)^{2}+1/(50\%)^{2})}=36\%$ , while in NoCor, the posterior uncertainty equals to 50%. In this situation, if the benchmark is chosen too high (e.g. 50%), it is too easy for ASS scenario, while it still requires a lot of constraints from satellite observations in NoCor scenario. If we choose 60% as the benchmark for assessing the posterior uncertainty, then the prior uncertainty in emissions in ASS will always below the

benchmark, even without conducting the inversion. Given different values of prior uncertainty in different scenarios, it is not easy to find a metric to fairly compare the results from different scenarios. We choose 20% as a benchmark because if the posterior uncertainty is below 20%, it is mainly determined by the projection of uncertainties in satellite observations on the uncertainty of emissions.

Furthermore, the posterior uncertainty in the emissions within 3 h time window or in the daily emissions, and thus the number of N20 and D20 are among the diagnostics we investigated on the potential of satellite observations. We also assessed the posterior uncertainty at annual scale, which integrates the uncertainty in all time windows, not only those whose uncertainty is smaller than 20%.

In the first version of this paper, we did consider to use a 2-D matrix to show the results, as shown in Fig. R2. We think such a 2-D matrix plot has its own disadvantages: 1) as stated above, the posterior uncertainty also depends on the prior uncertainty, if the threshold is chosen high, it does not properly represent the actual constraints from satellite observations; 2) such a plot cannot show the large variations in the number of cases within each emission bin. But this information is easy to read from the whisker plot in Fig. 3-5; and 3) such a 2-D matrix plot cannot compare the performance of the inversion in different experiments directly. Given the close values of some experiments (e.g. AMS and ASS in Fig. 3), the difference between experiments cannot be noticed by eye from separate 2-D matrix plots. Given these considerations, we decided to use the plots that have been shown in the paper, which can synthesize as the most information as we want to deliver, and also makes it possible to compare the performance for different experiments.

---

## Author Response (AR2)

**Response to comments on "PMIF v1.0: an inversion system to estimate the potential of satellite observations to monitor fossil fuel CO2 emissions" by Y. Wang et al.**

We thank the referee for reviewing our manuscript. Please find attached a point-by point reply (in black) to each of the comments raised by the referee (in blue). Changes in the revised manuscript are highlighted with dark red. All the pages and line numbers correspond to the original version of text.

The authors have addressed some of my comments but there are remaining issues that, in my opinion, need to be addressed.

First, as I mentioned in my initial review, the title is misleading. The work does not address fossil fuel emissions and claiming otherwise is disingenuous. Much of the work they cited used additional measurements to distinguish the sources. Further, this is a synthetic data study and, as the authors mention in their response, the package was developed to do OSSEs. It *can* be used for inversions, but that is not what was done. There are numerous confounding factors that need to be taken into account when doing flux inversions with real satellite data that have not been addressed here. I would strongly prefer a title that more accurately represents what was done. Something to the effect of: "PMIF v1.0: assessing the potential of satellite observations to constrain high resolution synthetic CO2 emissions over the globe".

**Response:**

We change the title to "PMIF v1.0: assessing the potential of satellite observations to constrain $CO_2$ emissions from large cities and point sources over the globe using synthetic data". In addition, we also revise the text to say we deal with $CO_2$ emissions from cities and power plants, not stressing fossil fuel emissions.

The explanation of what the authors are actually doing still needs improvement. In particular, the nomenclature of Section 2.7.2 must be improved. This is the section that explains much of what the algorithm is actually doing. This seems critical for publication in GMD, as GMD is focused on model development and I'm doubtful that readers will be able to follow this. It is currently difficult to even follow what the variables represent. Variables seem to arise out of nowhere and then don't seem to be used. For example, the definition of "a_{spt,i,j}(k)" and "b_{spt,i,j}(k)" seems to be the diagonal elements of "\mathbf{A}_{spt,i,j}(k)" and "\mathbf{B}_{spt,i,j}(k)", respectively, but I'm not actually sure from the text. Then "a_{spt,i,j}(k)" and "b_{spt,i,j}(k)" don't seem to be used after they are defined. I'd assumed the authors were using boldface capital variables to denote matrices, but Eq 8 suggests that "\mathbf{M}^T_{I,J,k}\hat{\mathbf{R}}^{-1}_{I,J,k}\mathbf{M}_{I,J,k}" is actually a scalar. Otherwise I'm not sure how the matrix on the right is formed. The authors sometimes have "k" as a subscript but other times it is in parentheses. All of this makes the work extremely difficult to follow. Giving more description to "\mathbf{M}^T_{I,J,k}\hat{\mathbf{R}}^{-1}_{I,J,k}\mathbf{M}_{I,J,k}" and consistent subscriptng would be useful.

**Response:**

We revise the paragraphs between Ln 391-408 to ensure that the bold capital letters are used for matrices and italic lower-case letters are used for scales:

"In a first step, Eq. (1) is applied to each $10^\circ \times 10^\circ$ spatial inversion windows on each day separately (corresponding to an 8:30-11:30 time window for clumps within the spatial inversion windows), by using

the corresponding blocks in **B**:

$$\mathbf{A}_{\mathrm{spt},p,q} = \left(\mathbf{B}_{\mathrm{spt},p,q}^{-1} + \mathbf{M}_{\mathrm{spt},p,q}^{\mathrm{T}}\mathbf{R}_{\mathrm{spt},p,q}^{-1}\mathbf{M}_{\mathrm{spt},p,q}\right)^{-1} \tag{6}$$

Where $p$ is the $p$th spatial inversion window and $q$ is the $q$th day during one year. Here, $\mathbf{B}_{\mathrm{spt},p,q}$ is a diagonal matrix that only contains the variances of prior uncertainties in emissions during 8:30-11:30 for the clumps within the spatial inversion window. $\mathbf{M}_{\mathrm{spt},i,j}$ accounts for the spatial overlap of plumes generated from nearby clumps. Then we derive an "instant" scalar to represent the observational constraint for a given clump $s$ in the 8:30-11:30 time window on day $q$ (denoted as $r_{q,s}$ hereafter):

$$r_{q,s}=1/a_{\mathrm{spt},q,s} - 1/b_{\mathrm{spt},q,s} \tag{7}$$

Where $a_{\mathrm{spt},q,s}$ is a scalar on the diagonal of $\mathbf{A}_{\mathrm{spt},p,q}$ representing the variance of posterior uncertainty of emission from clump $s$ in $p$th spatial inversion window and in 8:30-11:30 time window on day $q$ obtained by Eq. (6), and $b_{\mathrm{spt},q,s}$ is a scalar on the diagonal of $\mathbf{B}_{\mathrm{spt},p,q}$ representing the variance of prior uncertainty for the same control variable.

In the second step, the inversion is conducted for each clump separately, considering the correlation in time in the prior uncertainties, using $r_{q,s}$ derived from the first step:

$$\mathbf{A}_{\mathrm{tmp},s} = \left( \mathbf{B}_{\mathrm{tmp},s}^{-1} + \begin{bmatrix} r_{1,s} & 0 & 0 & 0 & \cdots & 0 & 0 \\ 0 & 0 & 0 & 0 & \cdots & 0 & 0 \\ 0 & 0 & r_{2,s} & 0 & \cdots & 0 & 0 \\ 0 & 0 & 0 & 0 & \cdots & 0 & 0 \\ \vdots & \vdots & \vdots & \vdots & \ddots & \vdots & \vdots \\ 0 & 0 & 0 & 0 & \cdots & r_{366,s} & 0 \\ 0 & 0 & 0 & 0 & \cdots & 0 & 0 \end{bmatrix} \right)^{-1} \tag{8}$$

Where $\mathbf{B}_{\mathrm{tmp},s}$ is the covariance matrix of the prior uncertainty for a given clump $s$ including the temporal auto-correlation:

$$\mathbf{B}_{\mathrm{tmp},s} = \begin{bmatrix} \sigma_{t1}^2 & cov(\varepsilon_{t1},\varepsilon_{t2}) & \cdots & cov(\varepsilon_{t1},\varepsilon_{tn}) \\ cov(\varepsilon_{t1},\varepsilon_{t2}) & \sigma_{t2}^2 & \cdots & cov(\varepsilon_{t2},\varepsilon_{tn}) \\ \vdots & \vdots & \ddots & \vdots \\ cov(\varepsilon_{t1},\varepsilon_{tn}) & cov(\varepsilon_{t2},\varepsilon_{tn}) & \cdots & \sigma_{tn}^2 \end{bmatrix} \tag{9}$$

Where n=366×2, corresponds to the number of time windows 8:30-11:30 and the rest 21 hours over the 366 days of one year (2008). t1, t3, etc. represent the 8:30-11:30 time windows and t2, t4, etc. represent the rest 21 hours.

In PMIF-Globe, we first…"

We also make some minor revisions in other parts of the manuscript, especially in the Method section.

Line 389: Authors should change "proven" to "show". Proven implies a mathematical proof, which has not been done.

**Response:**

We change the word "proven" to "shown" in this sentence.

[revised manuscript text omitted]

Where $i$ $p$ is the $i$th $p$th spatial inversion window and $j$ $q$ is the $j$th $q$th day during one year. Here, $\mathbf{B}_{\text{spt,pi,qj}}$ is a diagonal matrix that only contains the variances of prior uncertainties in emissions during 8:30-11:30 for the clumps within the spatial inversion window. $\mathbf{M}_{\text{spt,pi,qj}}$ accounts for the spatial overlap of plumes generated from nearby clumps. Then we derive an "instant" scalar to represent the observational constraint for a given clump $s$ in the 8:30-11:30 time window on day $q$ (denoted as $r_{q,s}$ hereafter)a "instant" $\mathbf{M}^{\text{T}}\mathbf{R}^{-1}\mathbf{M}$ (denoted as $\mathbf{M}_{i,j,k}^{\text{T}}\widehat{\mathbf{R}_{i,j,k}^{-1}}\mathbf{M}_{i,j,k}$) for a given clump $k$ at each 8:30-11:30 time window:

$$\mathbf{M}_{i,j,k}^{\text{T}}\widehat{\mathbf{R}_{i,j,k}^{-1}}\mathbf{M}_{i,j,k} = \left(\mathbf{A}_{\text{spt,i,j}}(k)^{-1} - \mathbf{B}_{\text{spt,i,j}}(k)^{-1}\right)^{-1} \tag{7}$$

$$r_{q,s} = 1/a_{\text{spt,q,s}} - 1/b_{\text{spt,q,s}} \tag{7}$$

Where $a_{\text{spt,q,s}}$ $a_{\text{spt,i,j}}(k)$ is a scalar on the diagonal offrom $\mathbf{A}_{\text{spt,pi,qj}}$ representing the variance of posterior uncertainty of emission from clump $k$ $s$ in $i$th $p$th spatial inversion window and in 8:30-11:30 time window on day $j$ $q$ obtained by Eq. (6), and $b_{\text{spt,q,s}}$ $b_{\text{spt,i,j}}(k)$ is thea scalar on the diagonal offrom $\mathbf{B}_{\text{spt,pi,qj}}$ representing the variance of prior uncertainty for the same control variable.

In the second step, the inversion is conducted for each clump $k$ $s$ separately, considering the correlation in time in the prior uncertaintiesB, using $r_{q,s}$ $\mathbf{M}_{i,j,k}^{\text{T}}\widehat{\mathbf{R}_{i,j,k}^{-1}}\mathbf{M}_{i,j,k}$ derived from the first step:

$$\mathbf{A}_{tmp,k} = \left(\mathbf{B}_{tmp,k}^{-1} + \begin{bmatrix} \mathbf{M}_{i,1,k}^{\text{T}}\widehat{\mathbf{R}_{i,1,k}^{-1}}\mathbf{M}_{i,1,k} & 0 & 0 \\ 0 & \ddots & 0 \\ 0 & 0 & \mathbf{M}_{
[revised manuscript text omitted]